# The half-life of the bone-derived hormone osteocalcin is regulated through *O*-glycosylation in mice, but not in humans

Omar Al Rifai[1,2], Catherine Julien[1], Julie Lacombe[1], Denis Faubert[3], Erandi Lira-Navarrete[4], Yoshiki Narimatsu[4], Henrik Clausen[4], Mathieu Ferron[1,2,5,6]*

[1]Molecular Physiology Research unit, Institut de Recherches Cliniques de Montréal, Montréal, Canada; [2]Programme de biologie moléculaire, Université de Montréal, Montréal, Canada; [3]Proteomics Discovery Platform, Institut de Recherches Cliniques de Montréal, Montréal, Canada; [4]University of Copenhagen, Faculty of Health Sciences, Copenhagen Center for Glycomics, Departments of Cellular and Molecular Medicine, Copenhagen, Denmark; [5]Département de Médecine, Université de Montréal, Montréal, Canada; [6]Division of Experimental Medicine, McGill University, Montréal, Canada

**Abstract** Osteocalcin (OCN) is an osteoblast-derived hormone with pleiotropic physiological functions. Like many peptide hormones, OCN is subjected to post-translational modifications (PTMs) which control its activity. Here, we uncover *O*-glycosylation as a novel PTM present on mouse OCN and occurring on a single serine (S8) independently of its carboxylation and endoproteolysis, two other PTMs regulating this hormone. We also show that *O*-glycosylation increases OCN half-life in plasma ex vivo and in the circulation in vivo. Remarkably, in human OCN (hOCN), the residue corresponding to S8 is a tyrosine (Y12), which is not *O*-glycosylated. Yet, the Y12S mutation is sufficient to *O*-glycosylate hOCN and to increase its half-life in plasma compared to wildtype hOCN. These findings reveal an important species difference in OCN regulation, which may explain why serum concentrations of OCN are higher in mouse than in human.

*For correspondence: mathieu.ferron@ircm.qc.ca

**Competing interests:** The authors declare that no competing interests exist.

## Introduction

Osteocalcin (OCN) is a peptide hormone secreted by osteoblasts, the bone forming cells (*Lee et al., 2007*). It regulates glucose metabolism by promoting beta cell proliferation and insulin secretion, and by improving insulin sensitivity (*Pi et al., 2011*; *Ferron et al., 2012*). In addition to its role in the regulation of energy metabolism, OCN is also involved in male fertility by promoting testosterone synthesis by Leydig cells (*Oury et al., 2011*), in muscle adaptation to exercise by improving glucose and fatty acid uptake in myocytes (*Mera et al., 2016a*), and in acute stress response through the inhibition of post-synaptic parasympathetic neurons (*Berger et al., 2019*). Overall, OCN might acts as an 'anti-geronic' circulating factor preventing age-related cognitive decline and muscle wasting (*Khrimian et al., 2017*; *Mera et al., 2016b*; *Oury et al., 2013b*). The G protein coupled receptor family C group six member A (GPRC6A) mediates OCN function in beta cells, muscles and testis (*Mera et al., 2016a*; *Pi et al., 2011*; *Oury et al., 2013a*), while the G protein coupled receptor 158 (Gpr158) mediates its function in the brain (*Khrimian et al., 2017*; *Kosmidis et al., 2018*).

Within the bone tissue, OCN undergoes a series of post-translational modifications (PTM) that are critical for the regulation of its endocrine functions. Prior to its secretion, in the osteoblast endo-plasmic reticulum, the OCN precursor (pro-OCN) is γ-carboxylated on three glutamic acid residues (Glu) by the vitamin K-dependent γ-glutamyl carboxylase (*Ferron et al., 2015*). In the trans-Golgi

**eLife digest** Bones provide support and protection for organs in the body. However, over the last 15 years researchers have discovered that bones also release chemicals known as hormones, which can travel to other parts of the body and cause an effect. The cells responsible for making bone, known as osteoblasts, produce a hormone called osteocalcin which communicates with a number of different organs, including the pancreas and brain.

When osteocalcin reaches the pancreas, it promotes the release of another hormone called insulin which helps regulate the levels of sugar in the blood. Osteocalcin also travels to other organs such as muscle, where it helps to degrade fats and sugars that can be converted into energy. It also has beneficial effects on the brain, and has been shown to aid memory and reduce depression.

Osteocalcin has largely been studied in mice where levels are five to ten times higher than in humans. But it is unclear why this difference exists or how it alters the role of osteocalcin in humans. To answer this question, Al Rifai et al. used a range of experimental techniques to compare the structure and activity of osteocalcin in mice and humans.

The experiments showed that mouse osteocalcin has a group of sugars attached to its protein structure, which prevent the hormone from being degraded by an enzyme in the blood. Human osteocalcin has a slightly different protein sequence and is therefore unable to bind to this sugar group. As a result, the osteocalcin molecules in humans are less stable and cannot last as long in the blood. Al Rifai et al. showed that when human osteocalcin was modified so the sugar group could attach, the hormone was able to stick around for much longer and reach higher levels when added to blood in the laboratory.

These findings show how osteocalcin differs between human and mice. Understanding this difference is important as the effects of osteocalcin mean this hormone can be used to treat diabetes and brain disorders. Furthermore, the results reveal how the stability of osteocalcin could be improved in humans, which could potentially enhance its therapeutic effect.

---

network, pro-OCN is next cleaved by the proprotein convertase furin releasing mature carboxylated OCN (Gla-OCN) (*Al Rifai et al., 2017*). The presence of the negatively charged Gla residues allows Gla-OCN to bind hydroxyapatite, the mineral component of the bone extracellular matrix (ECM). It is during bone resorption that Gla-OCN is decarboxylated through a non-enzymatic process involving the acidic pH generated by the osteoclasts, ultimately leading to the release of bioactive uncarboxylated OCN (ucOCN) in the circulation (*Ferron et al., 2010a*; *Lacombe et al., 2013*). The conclusion that ucOCN represents the bioactive form of this protein in rodents is supported by cell-based assays, mouse genetics and in vivo studies [reviewed in *Mera et al., 2018*].

The role of OCN in the regulation of glucose metabolism appears to be conserved in humans. Human ucOCN can bind and activate human GPRC6A (*De Toni et al., 2016*) and promotes beta cell proliferation and insulin synthesis in human islets (*Sabek et al., 2015*), while mutations or polymorphisms in human *GPRC6A* are associated with insulin resistance (*Di Nisio et al., 2017*; *Oury et al., 2013a*). Finally, several cross-sectional and observational studies have detected a negative association between OCN or ucOCN, and insulin resistance or the risk of developing type 2 diabetes in various human populations (*Lin et al., 2018*; *Turcotte et al., 2020*; *Lacombe et al., 2020*).

Yet, some important species divergences exist between mice and humans with regard to OCN biology. First, only 30 out of the 46 amino acids (i.e. 65%) composing mature mouse OCN are conserved in human OCN. This is in striking contrast with other peptide hormones involved in the control of energy metabolism such as leptin and insulin whose respective sequence display about 85% conservation between mouse and human. Second, the circulating concentrations of OCN, even though decreasing with age in both species, are five to ten times higher in mice than in humans throughout life span [(*Mera et al., 2016a*) see also *Table 1*]. Based on these observations, we hypothesized that the post-translational regulation of OCN may be different between these two species, resulting in increased mouse OCN half-life in circulation.

Here, using proteomics and cell-based assays, we identified *O*-glycosylation as a novel PTM presents in mouse OCN, and showed that this modification increases mouse OCN half-life in plasma ex vivo and in vivo. In contrast, mature human OCN does not contain the *O*-glycosylation site found

**Table 1.** OCN serum levels in mouse and human at different ages.

| Age (mice/human) | OCN serum levels (ng/ml) | |
| --- | --- | --- |
| | Mouse [mean ± SD (n)] | Human* [mean ± SD (n)] |
| 2 weeks/ 1 year old | 1369.7 ± 146.7 (8) | 62.9 ± 8.1 (43) |
| 4 weeks/ 11–13 years old | 617.2 ± 192.5 (5) | 74.1 ± 8.9 (41) |
| 13 weeks/ 25–29 years old | 252.2 ± 8.0 (4) | 21.0 ± 6.3 (49) |
| 60 weeks/ 50–54 years old | 50.0 ± 7.2 (4) | 13.5 ± 6.3 (127) |

*(*Hannemann et al., 2013*; *Cioffi et al., 1997*).

The online version of this article includes the following source data for Table 1:

Source data 1. Numerical data from *Table 1*.

in the mouse protein and consequently is not normally glycosylated. Yet, a single point mutation in human OCN is sufficient to elicit its *O*-glycosylation and to increase its half-life in plasma.

## Results

### Mouse OCN is *O*-glycosylated on a single serine residue

To better document the circulating level of OCN in humans and mice, we measured the serum concentration of OCN in wildtype mice at different ages (2 to 60 weeks) and compared the values with the reported serum level of OCN at corresponding life periods in humans (*Table 1*). This analysis reveals that serum OCN level is five- to ten-time lower in humans than in mice throughout life. One potential explanation for this observation could be that a mouse specific PTM increases mouse OCN half-life in circulation. Since OCN Gla residues and pro-OCN cleavage site are conserved between mouse and human, we searched for additional PTMs present in mouse OCN and characterized their impact on OCN half-life.

To that end, OCN was immunoprecipitated from the secretion medium of primary mouse osteoblast cultures or from mouse bone protein extracts using specific polyclonal goat antibodies recognizing OCN C-terminus sequence (*Ferron et al., 2010b*). OCN was then characterized without proteolysis by reverse-phase HPLC followed by mass spectrometry (MS) and tandem mass spectrometry (MS/MS). This 'top-down' analysis revealed that the most abundant OCN forms have a monoisotopic mass ranging from 5767.6961 to 6441.7636 Da which exceeds the predicted mass of 5243.45 Da corresponding to the fully carboxylated (3 × Gla) OCN (*Table 2* and *Table 3*). According to the various monoisotopic masses observed, we predict that this difference could be mainly explained by the presence of a single *O*-linked glycan adduct composed of one *N*-acetylgalactosamine (GalNAc), one galactose (Gal) and one or two *N*-acetylneuraminic acid (NANA). In addition to *O*-glycosylation, minor additional monoisotopic mass change corresponding to oxidation, unidentified modifications and/or adduct ions were also detected. Uncarboxylated OCN does not accumulate in bone ECM (*Ferron et al., 2015*) and accordingly, only fully or partially carboxylated OCN was detected in bone extracts, both of which were found to be *O*-glycosylated. In contrast, significant amount of ucOCN could be detected in osteoblast culture supernatants, most likely due to the low level of vitamin K present in fetal bovine serum. Importantly, in osteoblast supernatant the *O*-linked glycan adduct could be detected on both carboxylated and uncarboxylated OCN (*Table 2*).

Using multiple approaches, we next established that mouse OCN is indeed subjected to *O*-glycosylation in cells and in vivo. First, in SDS-PAGE analyses the apparent molecular weight of OCN is reduced when expressed in HEK293 cells where the O-glycosylation capacity has been engineered to truncate O-glycans by knockout of *C1GALT1C1* which encodes COSMC (core 1β3-Gal-T-specific molecular chaperone), a private chaperone required for the elongation of *O*-glycans (*Figure 1A*; *Steentoft et al., 2011*). Second, when expressed in CHO-ldlD cells, which have defective UDP-Gal/UDP-GalNAc 4-epimerase and are hence deficient in *O*-glycosylation (*Kingsley et al., 1986*), OCN apparent molecular weight is also reduced compared to the same protein expressed in the parental CHO cell line. Supplementation of Gal and GalNAc in the culture medium rescued the

**Table 2.** The monoisotopic mass and relative abundance of the different OCN forms detected in the supernatant of differentiated osteoblasts.

| Monoisotopic mass range (Da) | Relative abundance (%) | Most probable modification | Most probable oligosaccharide |
|---|---|---|---|
| O-glycosylated OCN | 83.88 | | |
| 5767.6961 | 4.80 | Glycosylation | HexNAc, Hex, NANA |
| 5783.6801 | 0.25 | Glycosylation + oxidation | HexNAc, Hex, NANA |
| 5855.6676 | 4.60 | Glycosylation + 2x Gla | HexNAc, Hex, NANA |
| 5899.7161 | 3.16 | Glycosylation + 3x Gla | HexNAc, Hex, NANA |
| 5915.6386–5968.5796 | 5.51 | Glycosylation + 3x Gla + oxidation + additional unidentified modifications or adduct ions | HexNAc, Hex, NANA |
| 6058.7916 | 24.48 | Glycosylation | HexNAc, Hex, 2x NANA |
| 6074.7766–6096.7409 | 2.48 | Glycosylation + oxidation | HexNAc, Hex, 2x NANA |
| 6102.7681 | 7.89 | Glycosylation + 1x Gla | HexNAc, Hex, 2x NANA |
| 6146.7609 | 23.97 | Glycosylation + 2x Gla | HexNAc, Hex, 2x NANA |
| 6190.8061–6214.7278 | 6.72 | Glycosylation + 3x Gla + additional unidentified modifications or adduct ions | HexNAc, Hex, 2x NANA |
| Non O-glycosylated OCN | 16.12 | | |
| 5127.4676 | 3.00 | Oxidation | NA |
| 5171.4301 | 1.85 | 1x Gla + oxidation | NA |
| 5199.4446 | 2.08 | 2x Gla | NA |
| 5215.4296 | 8.50 | 2x Gla + oxidation | NA |
| 5259.4204 | 0.66 | 3x Gla + oxidation | NA |

Gla: Gamma-carboxyglutamic acid residue; HexNAc: N-acetylhexosamine; Hex: Hexose; NANA: N-acetylneuraminic acid; 1x: one time; 2x: two times; 3x: three times; NA: not applicable.

The online version of this article includes the following source data for Table 2:

Source data 1. Raw proteomic data from **Table 2**.

O-glycosylation defect of CHO-ldlD and restored the molecular shift in the secreted OCN (**Figure 1B**). Third, treatment of primary osteoblasts with benzyl-N-acetyl-α-galactosaminide (Gal-NAc-bn), an inhibitor of N-acetylgalactosaminyltransferases (GalNAc-Ts), the enzymes responsible for initiating O-glycosylation, decreases the apparent molecular weight of OCN secreted in the medium (**Figure 1C**). Finally, treatment of mouse bone extracts with neuraminidase and O-glycosidase, which removes respectively NANA, and core 1 and core 3 O-linked disaccharides, also decreases the apparent molecular weight of endogenous OCN (**Figure 1D**).

We next aimed at identifying which OCN residue(s) is(are) O-glycosylated. Mature mouse OCN contains three serine (S) and three threonine (T) residues (**Figure 1E**), the two main types of amino acids on which O-glycosylation occurs (**Steentoft et al., 2013**). As expected, mutating all serine and threonine residues into alanine abrogates OCN glycosylation in primary mouse osteoblasts as assessed by SDS-PAGE (**Figure 1F**). Further mutagenesis studies revealed that the O-glycosylation site resides within the N-terminal part of the protein, that is on S5, S8 or T15 (**Figure 1F**). Single amino acid mutagenesis allowed the identification of S8 as the O-glycosylation site of OCN in osteoblasts (**Figure 1G**), a result consistent with the MS/MS analysis of OCN isolated from bone which also suggested that this residue is the O-glycosylation site (**Figure 1H**). Together, these results establish that mouse OCN is O-glycosylated on at least one serine residue in cell culture and in vivo.

## Several polypeptide N-acetylgalactosaminyltransferases (GalNAc-Ts) redundantly O-glycosylate OCN independently of its carboxylation and processing

Protein O-glycosylation is initiated by the transfer of a GalNAc to a serine or threonine residue, a reaction taking place in the Golgi and catalyzed by GalNAc-Ts, a family of enzymes comprising 19

**Table 3.** The monoisotopic mass and relative abundance of the different OCN forms detected in mouse bone homogenates.

| Monoisotopic mass range | Relative abundance (%) | Most probable modification | Most probable oligosaccharide |
|---|---|---|---|
| *O*-glycosylated OCN | 99.07 | | |
| 5855.6676 | 0.54 | Glycosylation + 2x Gla | HexNAc, Hex, NANA |
| 5899.7161 | 7.43 | Glycosylation + 3x Gla | HexNAc, Hex, NANA |
| 5915.6386–6135.6796 | 36.07 | Glycosylation + 3x Gla + oxidation + additional unidentified modifications or adduct ions | HexNAc, Hex, NANA |
| 6146.7609–6162.7991 | 5.42 | Glycosylation + 2x Gla + additional unidentified modifications | HexNAc, Hex, 2xNANA |
| 6190.8061 | 4.49 | Glycosylation + 3x Gla | HexNAc, Hex, 2xNANA |
| 6206.8016–6441.7636 | 45.12 | Glycosylation + 3x Gla + oxidation + additional unidentified modifications or adduct ions | HexNAc, Hex, 2xNANA |
| Non *O*-glycosylated OCN | 0.93 | | |
| 5259.4204 | 0.93 | 3x Gla + oxidation | NA |

Gla: Gamma-carboxyglutamic acid residue; HexNAc: *N*-acetylhexosamine; Hex: Hexose; NANA: *N*-acetylneuraminic acid; 1x: one time; 2x: two times; 3x: three times; NA: not applicable.

The online version of this article includes the following source data for Table 3:

Source data 1. Raw proteomic data from **Table 3**.

different members in mice (**Bennett et al., 2012**). Quantitative PCR on mRNA isolated from undifferentiated and differentiated primary mouse osteoblasts revealed that several GalNAc-Ts are expressed in this cell type, with *Galnt1* and *Galnt2* being the most strongly expressed ones (**Figure 2A**). We noticed that S8A mutation abrogates OCN *O*-glycosylation in HEK293 cells and in primary osteoblasts (**Figure 1G** and data not shown). GalNAc-T3 and its paralogue GalNAc-T6 are known to be expressed in HEK293 (**Narimatsu et al., 2019b**) and our data shows they are also expressed in primary mouse osteoblasts and induced during osteoblast differentiation. Although these observations suggest one or both of these enzymes may be involved in OCN *O*-glycosylation, the inactivation of *GALNT3* and/or *GALNT6* genes failed to alter OCN *O*-glycosylation in HEK293 (**Figure 2B**). Since *GALNT1* and *GALNT2* are also highly expressed in osteoblasts and induced during osteoblast differentiation, we inactivated these two genes in combination with *GALNT3*, and assess the impact on OCN *O*-glycosylation in HEK293 cells. This partially abolished OCN glycosylation (**Figure 2B**), suggesting that these three GalNAc-Ts are the primary isoenzymes that redundantly initiate the *O*-glycosylation of OCN.

Processing of pro-OCN by the proprotein convertase furin and its γ-carboxylation are two posttranslational modifications regulating OCN endocrine function (**Ferron et al., 2015**; **Al Rifai et al., 2017**). We therefore next aimed at testing whether OCN *O*-glycosylation can interfere with its γ-carboxylation or processing, or inversely, if *O*-glycosylation is modulated by γ-carboxylation or processing of pro-OCN.

Pharmacological inhibition of γ-carboxylation or furin, using warfarin or Dec-RVKR-CMK (RVKR) respectively, did not impact OCN *O*-glycosylation in primary osteoblasts and HEK293 cells (**Figure 2C** and **Figure 2—figure supplement 1**). Similarly, inhibition of OCN *O*-glycosylation through GalNAc-bn treatment or the S8A mutation did not significantly affect its processing or its γ-carboxylation (**Figure 2C–E** and **Figure 2—figure supplement 1**). We also tested whether OCN processing influences its *O*-glycosylation in vivo. As shown in **Figure 2F**, both mature OCN present in control bones and pro-OCN present in furin-deficient bones are de-glycosylated by neuraminidase and *O*-glycosidase, indicating that pro-OCN is normally *O*-glycosylated in absence of processing by furin in vivo. Altogether, these results support the notion that osteocalcin *O*-glycosylation is not affected by its carboxylation status or by its processing by furin. Moreover, blocking *O*-glycosylation does not prevent pro-OCN processing by furin or its carboxylation.

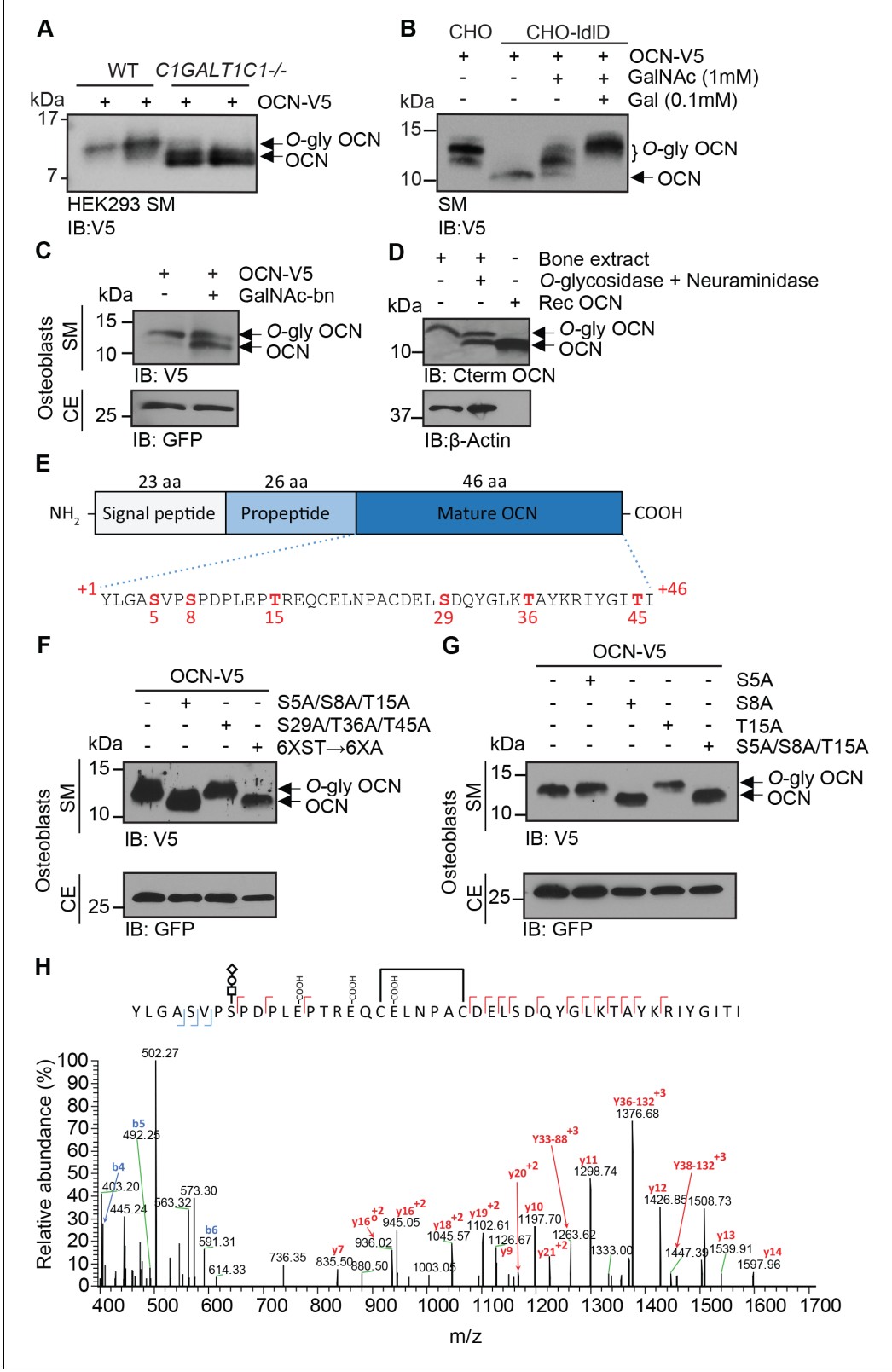

**Figure 1.** OCN is O-glycosylated in vitro and in vivo on serine 8. (**A**) Western blot analysis on the secretion medium (SM) of HEK293 (WT) and HEK293 lacking COSMC (*C1GALT1C1-/-*) transfected with mouse OCN-V5. (**B**) Western blot analysis on the SM of CHO and CHO-ldlD cells transfected with mouse OCN-V5. CHO-ldlD cells were treated or not with 0.1 mM Galactose (Gal) and/or 1 mM N-acetylgalactosamine (GalNAc). (**C**) Effect of *N-acetylgalactosaminyltransferase* (GalNAc-Ts) inhibition on mouse OCN O-glycosylation in osteoblasts. Western blot analysis on the SM and cell extract

*Figure 1 continued on next page*

*Figure 1 continued*

(CE) of primary osteoblasts transfected with mouse OCN-V5 and treated or not with 2 mM of GalNAc-bn. (D) OCN deglycosylation assay. Bone extracts of C57BL/6J mice were treated or not with O-glycosidase and neuraminidase for 4 hr at 37°C and analyzed by western blot using anti-C-terminal OCN antibody (Cterm OCN). β-actin was used as a loading control. Rec OCN: Non-glycosylated OCN produced in bacteria. (E) Structure of mouse pre-pro-OCN and amino acid sequence of mature mouse OCN. The serine (S) and threonine (S) residues are in red. (F) Western blot analysis on the SM and cell extract (CE) of primary osteoblasts transfected with OCN-V5 containing or not the indicated mutations. In the 6XST→6XA mutant, all six serine and threonine residues from OCN were mutated to alanine. (G) Western blot analysis on the SM and cell extract (CE) of primary osteoblasts transfected with OCN-V5 containing or not the indicated mutations. (H) Annotated HCD MS/MS spectrum of a modified form of OCN (HexNAc-Hex-NANA + 3 Gla + S-S) pulled down from the bone homogenate of C57BL/6J mice. The precursor m/z value is 1180.95003 (M+5H)$^{+5}$ and mass accuracy with the annotated OCN modified form is 4.6 ppm. In C, F and G, GFP co-expressed from OCN-V5 expression vector, was used as a loading control.

The online version of this article includes the following source data for figure 1:

**Source data 1.** Original western blot image from *Figure 1A*.
**Source data 2.** Original western blot image from *Figure 1B*.
**Source data 3.** Original western blot image from *Figure 1C*.
**Source data 4.** Original western blot image from *Figure 1D*.
**Source data 5.** Original western blot image from *Figure 1F*.
**Source data 6.** Original western blot image from *Figure 1G*.
**Source data 7.** Raw proteomic data from *Figure 1H*.

## *O*-glycosylation increases mouse OCN half-life in plasma ex vivo by preventing plasmin-mediated endoproteolysis

The results presented above suggest that *O*-glycosylation is not regulating the processing of pro-OCN by furin or the secretion of mature OCN by osteoblasts. It was recently observed that *O*-glycosylation can also increase the stability of some peptide hormones in the circulation by preventing proteolytic degradation (*Hansen et al., 2019*; *Madsen et al., 2020*). We therefore aimed at testing the impact of *O*-glycosylation on OCN half-life in plasma. To that end, we produced and purified *O*-glycosylated ucOCN from HEK293 and first compared its purity and molecular weight to native non-glycosylated ucOCN produced in bacteria by LC-MS, LC-MS/MS and SDS-PAGE (*Figure 3A,B*; *Figure 3—figure supplement 1*). Importantly, we observed that >99% of the ucOCN purified from HEK293 is O-glycosylated, with a certain proportion (~30%) containing two glycan adducts (*Figure 3A*), suggesting that in this context O-glycosylation may occurs on more than one residue. Freshly isolated *Bglap*-/- plasma, which is depleted of endogenous OCN, was next used to assess ucOCN half-life ex vivo. In all the following experiments, concentrations of ucOCN were measured with an ELISA assay recognizing both non-glycosylated and glycosylated mouse ucOCN (see Materials and methods and *Figure 3—figure supplement 2*). Non-glycosylated ucOCN has a half-life of ~120 min when incubated in plasma at 37°C, while *O*-glycosylated ucOCN is stable for more than 5 hr in the same conditions (*Figure 3C*). Non-glycosylated ucOCN was stable when incubated in a saline solution containing 3.5% BSA at 37°C for 2 hr (*Figure 3—figure supplement 3*), implying that ucOCN is not intrinsically unstable. In addition, stability of the non-glycosylated ucOCN was restored when incubated in plasma at 4°C or in heat-inactivated (HI) plasma at 37°C (*Figure 3D*), suggesting that non-glycosylated ucOCN's decline involves the action of a protease. Pepstatin A (Pep A) an aspartic proteases inhibitor, RVKR a proprotein convertases inhibitor and ethylenediaminetetraacetic acid (EDTA) which inhibits metalloproteases did not affect non-glycosylated ucOCN stability in plasma ex vivo (*Figure 3—figure supplement 3*). The OCN sequence surrounding S8 contains several proline residues (*Figure 1E*) and could therefore be recognized by prolyl endopeptidases, such as the fibroblast activation protein (FAP) which is present in the circulation (*Sánchez-Garrido et al., 2016*; *Coutts et al., 1996*). We thus also tested the effect of talabostat, an inhibitor of FAP and dipeptidyl peptidases, on ucOCN stability in plasma. Surprisingly, treatment of plasma with 10 mM talabostat did not inhibit, but rather increased non-glycosylated ucOCN degradation (*Figure 3E*). A reduced stability of non-glycosylated OCN in plasma was also observed in the presence of PMSF (*Figure 3F*) at a concentration (10 mM) that was also shown to inhibit dipeptidyl peptidases and prolyl endopeptidases (*Banbula et al., 2000*; *Bermpohl et al., 1998*). One function of FAP in vivo is to activate the α2-antiplasmin precursor releasing the active α2-antiplasmin, which in turn acts as an inhibitor of plasmin activity (*Lee et al., 2011*; *Lee et al., 2006*). Interestingly, mouse OCN contains

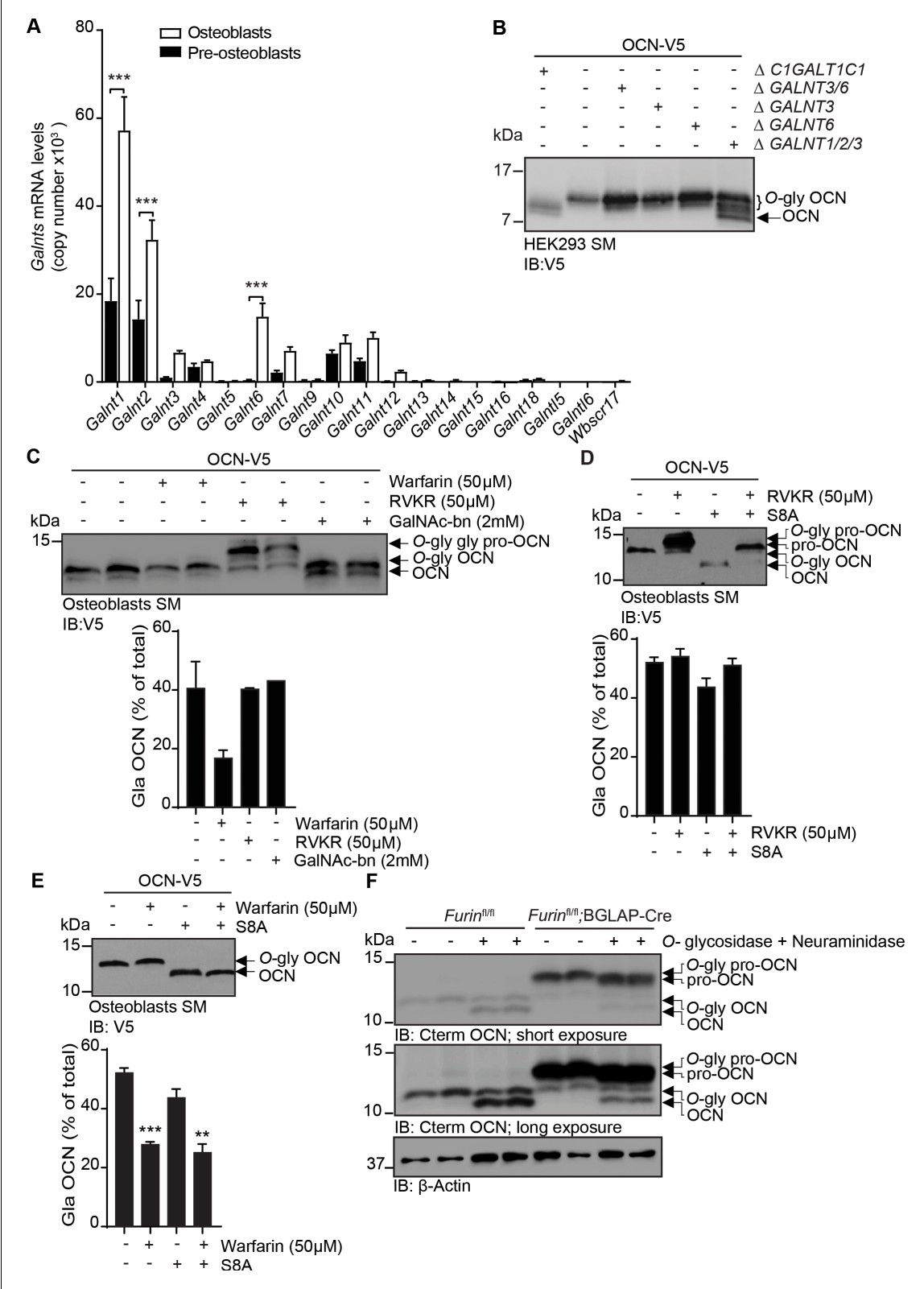

**Figure 2.** OCN O-glycosylation by N-*acetylgalactosaminyltransferase* (GalNAc-Ts) is independent of its processing and γ-carboxylation. (**A**) *Galnts* expression in pre-osteoblasts (undifferentiated) and osteoblasts (differentiated) by quantitative PCR (n = 3 per condition). Results are represented as copy number of *Galnts* normalized to *Actb*. (**B**) Western blot analysis of OCN in the secretion media (SM) of HEK293 cells deficient for specific GalNAc-Ts. OCN-V5 was transfected in parental, *C1GALT1C1-/-* (Δ *C1GALT1C1*), or GALNTs deficient (Δ) HEK293 cells and analysed by western blot using anti-

*Figure 2 continued on next page*

*Figure 2 continued*

V5 antibody. (C) Western blot analysis on the SM of osteoblasts transfected with mouse OCN-V5 and treated or not with 2 mM of GalNAc-bn, 50 µM warfarin or 50 µM Dec-RVKR-CMK (RVKR) (upper panel), and percentage of carboxylated OCN (Gla-OCN) over total OCN measured by ELISA (lower panel; n = 2 per condition). (D) Western blot analysis on the SM of osteoblasts transfected with mouse OCN-V5 containing or not the S8A mutation and treated with 50 µM Dec-RVKR-CMK (RVKR) (upper panel), and percentage of carboxylated OCN (Gla-OCN) over total OCN measured by ELISA (lower panel; n = 3 per condition). (E) Western blot analysis on the SM of osteoblasts transfected with mouse OCN-V5 containing or not the S8A mutation and treated with 50 µM warfarin (upper panel), and percentage of carboxylated OCN over total OCN measured by ELISA (lower panel; n = 3 per condition). (F) Western blot analysis of OCN deglycosylation assay on bone extracts from *Furin*[fl/fl] and *Furin*[fl/fl];BGLAP-Cre mice (n = 2 independent mice per genotype). Bone extracts were treated or not with O-glycosidase and neuraminidase for 4 hr at 37˚C and analyzed by western blot using anti-C-termimal OCN antibody (Cterm OCN). **p<0.01; ***p<0.001 using one-way ANOVA with Bonferroni multiple comparisons test.

The online version of this article includes the following source data and figure supplement(s) for figure 2:

**Source data 1.** Numerical data from the graph in *Figure 2A*.
**Source data 2.** Original western blot image from *Figure 2B*.
**Source data 3.** Original western blot image from *Figure 2C*.
**Source data 4.** Numerical data from the graph in *Figure 2C*.
**Source data 5.** Original western blot image from *Figure 2D*.
**Source data 6.** Numerical data from the graph in *Figure 2D*.
**Source data 7.** Original western blot image from *Figure 2E*.
**Source data 8.** Numerical data from the graph in *Figure 2E*.
**Source data 9.** Original western blot image from *Figure 2F*.
**Figure supplement 1.** Mouse OCN O-glycosylation occurs independently of its carboxylation and processing in HEK293 cells.

arginine residues in its N- and C-terminus (R16 and R40, respectively), which could be potential cleavage sites for plasmin (*Rawlings et al., 2008*). Together these observations led us to hypothesize that plasmin could be a protease responsible for ucOCN degradation in plasma. Supporting this notion, ucOCN is rapidly degraded when low concentration of recombinant plasmin is added to previously heat-inactivated plasma or in Tris buffered solution (*Figure 3G* and data not shown). Moreover, O-glycosylation partially protect ucOCN from plasmin-mediated degradation in the same assay (*Figure 3G*). Altogether, these data indicate that O-glycosylation protects ucOCN from plasmin mediated proteolysis, thereby increasing its half-life in plasma in vitro.

## Glycosylation increases mouse OCN stability in vivo

We next examined the stability of glycosylated and non-glycosylated mouse ucOCN in vivo by injecting an equal dose of each of these proteins in *Bglap-/-* mice which are depleted of endogenous OCN. In fasted animals, following an injection of 40 ng/g of body weight of ucOCN, the level of glycosylated ucOCN remains higher compared to the non-glycosylated form for the following 90 min (*Figure 3H*). Moreover, when expressed as a percentage of the maximum concentration reached at 30 min, the concentration of non-glycosylated OCN declines more rapidly than the one of glycosylated ucOCN (*Figure 3I*). Based on these curves, we estimated the in vivo half-life of O-glycosylated and non-glycosylated ucOCN to ~182 and~108 min respectively. In fed mice, glycosylated ucOCN serum concentration also remains higher than the one of non-glycosylated ucOCN for up to 2 hr following an injection of 40 ng/g of body weight (*Figure 3—figure supplement 4*). Circulating level of glycosylated ucOCN after 2 hr was further increased when 80 ng/g of body weight of protein was injected, while non-glycosylated ucOCN was not significantly increased with this higher dose (*Figure 3—figure supplement 4*). These results establish that O-glycosylation increases the stability of mouse OCN protein in vivo.

## Impact of glycosylation on OCN activity in culture

Using a cell-based assay, we next tested if O-glycosylation impacts the biological activity of OCN. INS-1 832/3 cells, a sub-clone of the INS-1 rat insulinoma cell line previously shown to express GPRC6A and to respond to OCN (*Ferron et al., 2010a*; *Pi et al., 2016*), were treated with vehicle or low doses of non-glycosylated and glycosylated ucOCN for 8 hr. At the end of the stimulation period, the expression of *Ins1*, the gene encoding pro-insulin 1, was assessed by quantitative PCR as a readout of OCN activity. As shown in *Figure 4A*, non-glycosylated ucOCN stimulation at 0.3 ng/ml could increase *Ins1* expression by 1.5-fold. However, a lower dose of glycosylated ucOCN (0.1 ng/

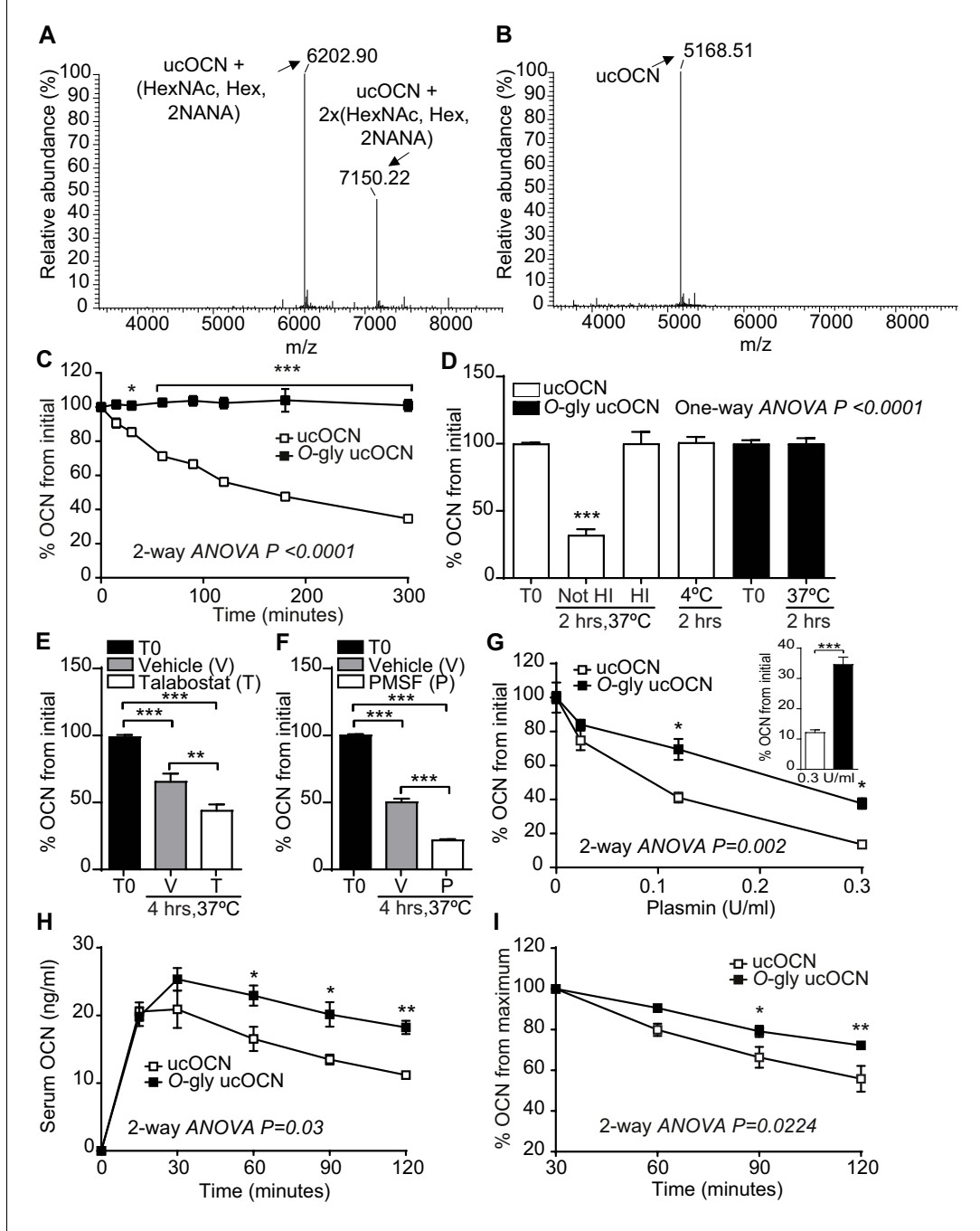

**Figure 3.** Mouse OCN O-glycosylation increases its half-life ex vivo and in vivo. (**A**) Annotated and deconvoluted MS spectrum of purified glycosylated mouse OCN (O-gly ucOCN). (**B**) Annotated and deconvoluted MS spectrum of purified non-glycosylated mouse OCN (ucOCN). (**C–D**) Ex vivo half-life of O-gly ucOCN and ucOCN in OCN deficient (*Bglap-/-*) plasma. (**C**) 100 ng/ml of O-gly ucOCN or ucOCN were incubated in plasma at 37˚C for 0 to 5 hr and OCN levels were measured at the indicated times (n = 4 independent plasma per condition). (**D**) 100 ng/ml of O-gly ucOCN or ucOCN were incubated in plasma for 2 hr at 37˚C in different conditions or at 4˚C. HI: heat-inactivated plasma (n = 3 independent plasma per condition). (**E**) 50 ng/ml of ucOCN was incubated in plasma from *Bglap-/-* mice for 4 hr at 37˚C in the presence of vehicle (**V**) or 10 mM talabostat (**T**) (n = 3 per condition). (**F**) 50 ng/ml of ucOCN was incubated in plasma from *Bglap-/-* mice for 4 hr at 37˚C in the presence of vehicle (**V**) or 10 mM phenylmethylsulfonyl fluoride (PMSF) (**P**) (n = 3 per condition). (**G**) Effect of plasmin on OCN stability ex vivo. 50 ng/ml of O-gly ucOCN and ucOCN were incubated for two hours in *Bglap-/-* heat-inactivated plasma containing different concentration of plasmin (n = 2 per condition). Inset graph shows the stability of the O-gly ucOCN and ucOCN incubated for two hours with 0.3 U/ml of plasmin (n = 5). (**H**) OCN deficient male mice (*Bglap-/-*) were fasted for 16 hr, O-gly ucOCN (n = 5 mice) or ucOCN (n = 5 mice) were injected intraperitoneally at a dose of 40 ng/g of body weight and serum OCN levels were measured at the indicated time points. (**I**) Using the data in (**H**) the percentage (%) of OCN in the declining phase was calculated relative to the maxima of each OCN

*Figure 3 continued on next page*

*Figure 3 continued*

forms at 30 min. T0: start point, see Materials and methods; HexNAc: N-acetylhexosamine; Hex: Hexose; NANA: N-acetylneuraminic acid. OCN measurements were performed using total mouse OCN ELISA assay (see Materials and methods). Results are given as mean ± SEM. *p<0.05; **p<0.01; ***p<0.001 using two-way ANOVA for repeated measurements with Bonferroni multiple comparisons test.

The online version of this article includes the following source data and figure supplement(s) for figure 3:

**Source data 1.** Raw proteomic data from *Figure 3A*.
**Source data 2.** Raw proteomic data from *Figure 3B*.
**Source data 3.** Numerical data from the graph in *Figure 3C*.
**Source data 4.** Numerical data from the graph in *Figure 3D*.
**Source data 5.** Numerical data from the graph in *Figure 3E*.
**Source data 6.** Numerical data from the graph in *Figure 3F*.
**Source data 7.** Numerical data from the graph in *Figure 3G*.
**Source data 8.** Numerical data from the graph in *Figure 3H*.
**Source data 9.** Numerical data from the graph in *Figure 3I*.
**Figure supplement 1.** Purification of recombinant O-glycosylated mouse ucOCN.
**Figure supplement 1—source data 1.** Original gel image from *Figure 3—figure supplement 1* (panel B).
**Figure supplement 2.** Mouse OCN ELISA recognize O-glycosylated mouse ucOCN (O-gly ucOCN) and non-glycosylated mouse ucOCN (ucOCN).
**Figure supplement 2—source data 1.** Numerical data from the graph in *Figure 3—figure supplement 2*.
**Figure supplement 3.** Effect of different protease inhibitors on non-glycosylated mouse ucOCN plasma half-life.
**Figure supplement 3—source data 1.** Numerical data from the graph in *Figure 3—figure supplement 3* (panel A).
**Figure supplement 3—source data 2.** Numerical data from the graph in *Figure 3—figure supplement 3* (panel B).
**Figure supplement 4.** Mouse OCN O-glycosylation increases its stability in vivo in fed conditions.
**Figure supplement 4—source data 1.** Numerical data from the graph in *Figure 3—figure supplement 4* (panel A).
**Figure supplement 4—source data 2.** Numerical data from the graph in *Figure 3—figure supplement 4* (panel B).

ml) was sufficient to significantly increase the expression of *Ins1* as compared to vehicle or to 0.3 ng/ml non-glycosylated ucOCN stimulation. These difference in biological activity could be explained at least in part by an increased stability of glycosylated ucOCN as compared to non-glycosylated ucOCN in INS-1 832/3 cell cultures (*Figure 4B*). Overall, these results suggest that glycosylated ucOCN is biologically active, at least in the setting of this cell-based assay, and that *O*-glycosylation also increases the stability of ucOCN in cell culture.

## Human OCN is not glycosylated

Sequence alignments revealed that the residue corresponding to S8 in the mouse protein is a tyrosine (Y12) in human OCN (*Figure 5A*). In addition, human OCN does not contain any serine or threonine residues and migrates at a lower molecular weight compared to mouse OCN when expressed and secreted by osteoblasts, HEK293 or CHO cells (*Figure 5B* and data not shown). Since mouse and human ucOCN have a very similar predicted molecular weight, that is 5.1 and 5.8 kDa, respectively, these observations suggested that human OCN may not be *O*-glycosylated. Remarkably, introduction of a single serine residue (Y12S mutation) in the human protein is sufficient to induce its *O*-glycosylation in osteoblasts as visualized by western blot (*Figure 5C*). In contrast, introducing a leucine at the same position (Y12L) did not alter human OCN apparent molecular weight, indicating that this tyrosine residue is not normally subjected to *O*-glycosylation. Since the apparent molecular weight of both native and Y12S human OCN are increased following treatment with RVKR, we concluded that *O*-glycosylation does not affect human OCN processing by furin (*Figure 5D*). These results establish that mature human OCN is not normally subjected to *O*-glycosylation, but that a single amino acid change (Y12S) is sufficient to induce its *O*-glycosylation in osteoblasts.

Because *O*-glycosylation impacts mouse ucOCN half-life in plasma, we next tested whether this PTM had a similar effect on human ucOCN. We produced and purified *O*-glycosylated human ucOCN[Y12S] from HEK293 and compared its purity and molecular weight to native non-glycosylated human ucOCN by LC-MS, LC-MS/MS and SDS-PAGE (*Figure 5E and F*, and *Figure 5—figure supplement 1*). Confirming what was observed in osteoblasts, human ucOCN containing the Y12S mutation purified from HEK293 was found to be *O*-glycosylated (*Figure 5E*). These proteins were then incubated in *Bglap-/-* mouse plasma at 37°C and the concentration of ucOCN monitored over time using a specific ELISA assay (*Lacombe et al., 2020*), which can quantify both non-glycosylated

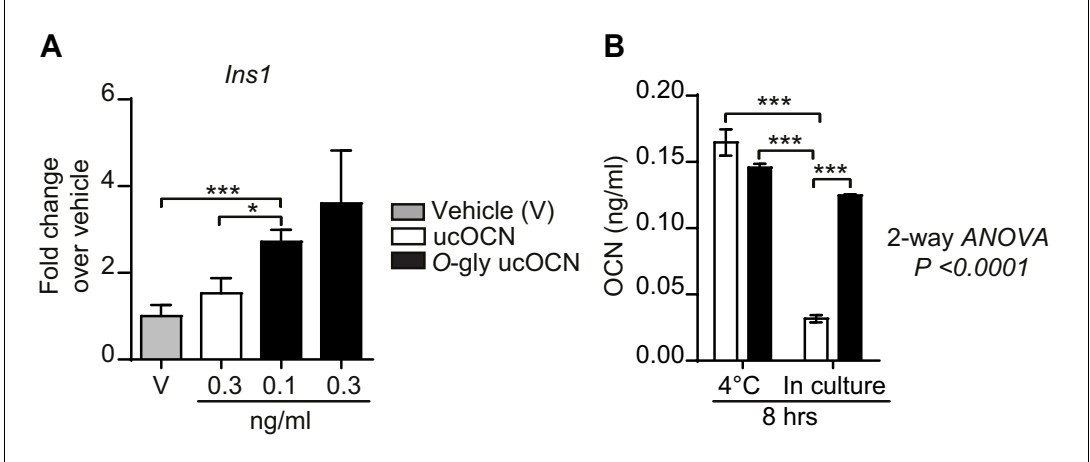

**Figure 4.** Effect of O-glycosylation on mouse OCN bioactivity and stability in culture. (**A**) Insulin gene expression (*Ins1*) in INS-1 832/3 cells following an 8 hr treatment with vehicle (n = 6), non-glycosylated mouse OCN (ucOCN) (n = 10) or glycosylated mouse ucOCN (O-gly ucOCN) (n = 8–10) at the indicated concentrations. (**B**) Concentration of OCN in the media incubated for 8 hr at 4°C without cells or for 8 hr in culture with INS-1 832/3 cells at 37°C (n = 3 for each condition). Results are shown as mean ± SEM. *p<0.05; ***p<0.001 using Student t-test or ordinary two-way ANOVA with Bonferroni multiple comparisons test.

The online version of this article includes the following source data for figure 4:

**Source data 1.** Numerical data from the graph in *Figure 4A*.
**Source data 2.** Numerical data from the graph in *Figure 4B*.

and *O*-glycosylated human ucOCN (see *Figure 5—figure supplement 2* and Materials and method). As shown in *Figure 5G*, in the conditions of this assay, non-glycosylated human ucOCN level declines by 50% within 180 min, while the concentration of the *O*-glycosylated version remains stable over the course of the experiment (i.e. 5 hr). As observed with the mouse protein, human ucOCN degradation was only inhibited when the plasma was heat-inactivated or incubated at 4°C (*Figure 5H* and *Figure 5—figure supplement 3*), suggesting that glycosylation protects mouse and human ucOCN from degradation through a similar mechanism.

## Discussion

In this study we identified *O*-glycosylation as a novel PTM regulating mouse ucOCN half-life in the circulation. We also showed that *O*-glycosylation is not found on human mature OCN, but that *O*-glycosylation of human OCN by means of a single amino acid change can improve its half-life in plasma ex vivo. These findings reveal an important species difference in the regulation of OCN and may also have important implication for the future use of recombinant ucOCN as a therapeutic agent in humans.

Numerous secreted proteins are subjected to mucin-type (GalNAc-type) *O*-glycosylation, a PTM which is initiated in the Golgi apparatus and involves multiple sequential glycosylation steps to produce diverse *O*-glycan structures (*Steentoft et al., 2013*; *Bennett et al., 2012*). The initiation of *O*-glycosylation is catalyzed by the GalNAc-Ts isoenzymes, however, the specific protein sequence(s) targeted by each of the GalNAc-Ts remain poorly characterized, although weak acceptor motifs for these have been identified by in vitro analyses (*Perrine et al., 2009*; *Gerken et al., 2006*). Here, we demonstrate that mouse OCN is *O*-glycosylated likely on serine 8, which is located within the amino acid sequence SVP[S]PDP[11]. Interestingly, this sequence strongly matches the consensus site previously defined for GalNAc-T1 and GalNAc-T2 (*Gerken et al., 2006*). In particular, the presence of proline residues in position −one, +one and +three has been shown to be determinant in the recognition of peptide substrates by GalNAc-T1 and GalNAc-T2 in vitro. We used an isogenic cell library with combinatorial engineering of isoenzyme families to explore the regulation of osteocalcin *O*-glycosylation by GalNAc-Ts (*Narimatsu et al., 2019a*; *Narimatsu et al., 2019b*). Combined knock out

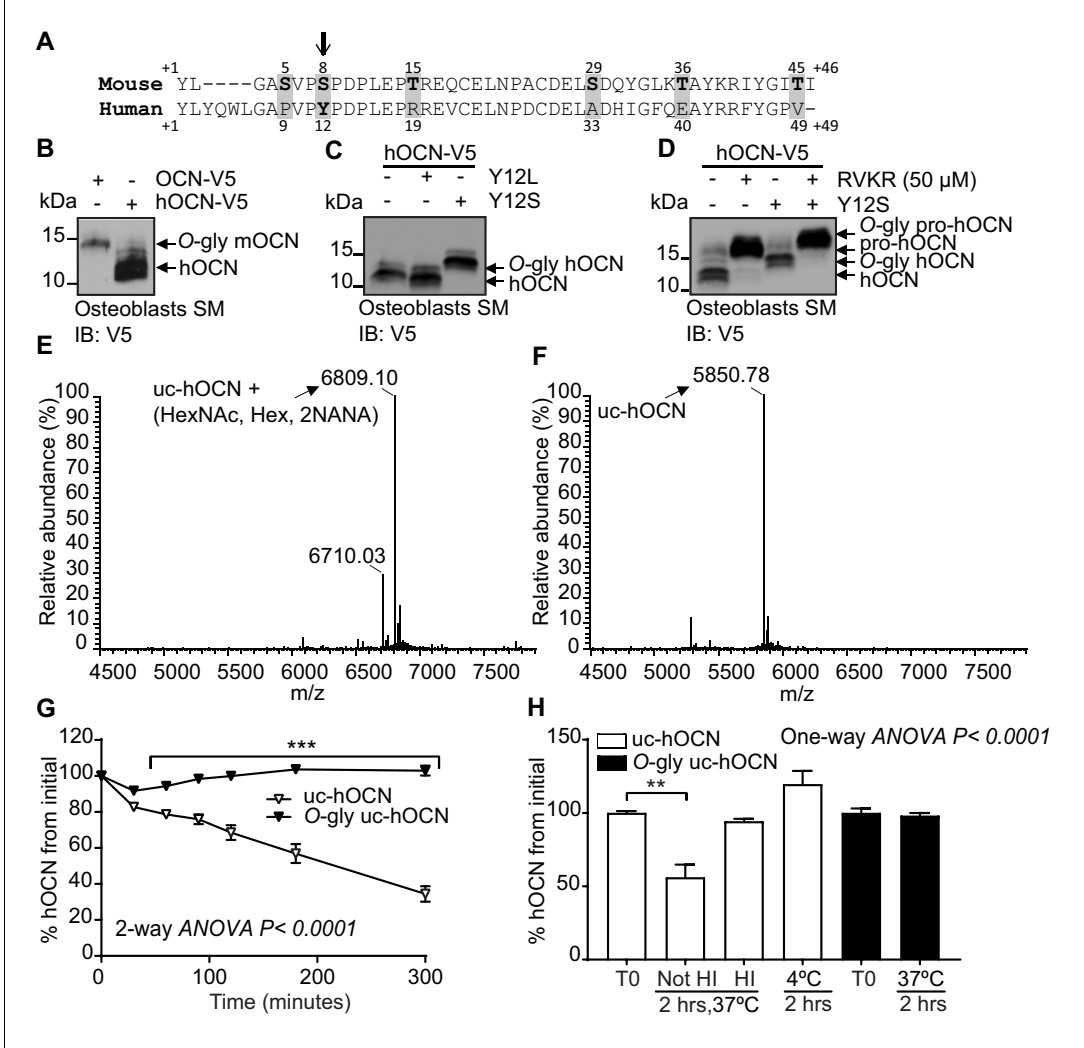

**Figure 5.** Human OCN O-glycosylation increases its half-life ex vivo. (**A**) Amino acid alignment of mouse and human OCN. The six serine and threonine residues present in the mouse protein and their corresponding amino acids in human OCN are highlighted in gray. The site of O-glycosylation in mouse OCN (S8) is indicated by an arrow. (**B**) Western Blot analysis on the secretion media (SM) of primary osteoblasts transfected with human OCN-V5 (hOCN) and mouse OCN-V5 (OCN). (**C**) Western blot analysis on the SM of primary osteoblasts transfected with hOCN-V5 containing or not the indicated mutations. (**D**) Western Blot analysis on the SM of primary osteoblasts transfected with hOCN-V5 containing or not the Y12S mutations and treated or not with 50 µM Dec-RVKR-CMK (RVKR). (**E**) Annotated and deconvoluted MS spectrum of purified O-glycosylated uncarboxylated human OCN (O-gly uc-hOCN). (**F**) Annotated and deconvoluted MS spectrum of and purified non-glycosylated uncarboxylated human OCN (uc-hOCN). (**G–H**) Ex vivo half-life of O-gly uc-hOCN and uc-hOCN in mouse plasma. (**G**) 60 ng/ml of O-gly uc-hOCN and uc-hOCN were incubated at 37°C in plasma of OCN deficient mice (*Bglap-/-*) (n = 4) for 0 to 5 hr and hOCN levels were measured at the indicated time. (**H**) O-gly uc-hOCN and uc-hOCN were incubated in *Bglap-/-* plasma for 2 hr at 37°C in different conditions or at 4°C (n = 4 independent plasma per condition). T0: start point, see Materials and methods; HI: heat-inactivated plasma; HexNAc: N-acetylhexosamine; Hex: Hexose; NANA: N-acetylneuraminic acid. Uc-hOCN levels were measured at the indicated time points using an uc-hOCN ELISA assay. Results are given as mean ± SEM. **p<0.01; ***p<0.001 using two-way ANOVA for repeated measurements with Bonferroni multiple comparisons test.

The online version of this article includes the following source data and figure supplement(s) for figure 5:

**Source data 1.** Original western blot image from *Figure 5B*.
**Source data 2.** Original western blot image from *Figure 5C*.
**Source data 3.** Original western blot image from *Figure 5D*.
**Source data 4.** Raw proteomic data from *Figure 5E*.
**Source data 5.** Raw proteomic data from *Figure 5F*.
**Source data 6.** Numerical data from the graph in *Figure 5G*.
**Source data 7.** Numerical data from the graph in *Figure 5H*.
**Figure supplement 1.** Purification of recombinant O-glycosylated human ucOCN.

*Figure 5 continued on next page*

*Figure 5 continued*

**Figure supplement 1—source data 1.** Original gel image from *Figure 5—figure supplement 1* (panel B).

**Figure supplement 2.** Human ucOCN ELISA equally recognize O-glycosylated human ucOCN (O-gly uc-hOCN) and non-glycosylated human ucOCN (uc-hOCN).

**Figure supplement 2—source data 1.** Numerical data from the graph in *Figure 5—figure supplement 2*.

**Figure supplement 3.** Effect of different protease inhibitors on non-glycosylated human ucOCN plasma half-life.

**Figure supplement 3—source data 1.** Numerical data from the graph in *Figure 5—figure supplement 3*.

of GalNAc-T1, 2 and 3 only partially abolishes mouse OCN *O*-glycosylation in HEK293 cells, suggesting that OCN may be a substrate for additional GalNAc-Ts.

*O*-glycosylation was shown to interfere with the action of proprotein convertases on several prohormones and receptors (*Goth et al., 2015*; *Kato et al., 2006*; *May et al., 2003*; *Schjoldager and Clausen, 2012*; *Schjoldager et al., 2010*; *Goth et al., 2017*). This appears not to be the case for OCN as its *O*-glycosylation does not interfere with its processing by furin in vitro and in vivo. In other proteins, such as leptin and erythropoietin, glycosylation adducts were shown to increase protein stability and half-life in circulation (*Elliott et al., 2003*; *Creus et al., 2001*). More recently, O-glycosylation was shown to protect atrial natriuretic peptide from proteolytic degradation by neprilysin or insulin-degrading enzyme in vitro (*Hansen et al., 2019*). Additional studies showed that sialic acid residues present in the glycosylation adducts increase protein charge, thereby improving serum half-life and decreasing liver and renal clearance (*Morell et al., 1971*; *Runkel et al., 1998*; *Perlman et al., 2003*; *Ziltener et al., 1994*). Although our data suggest that protection from proteolytic cleavage by plasmin in the plasma might be one mechanism by which *O*-glycosylation extend OCN half-life, we cannot exclude that in vivo, *O*-glycosylation may also decrease liver and renal clearance of OCN.

Mouse OCN contains arginine (R) residues at position 16 and 40, corresponding to R20 and R44 in human OCN, which could be potential cleavage sites for plasmin (*Figure 5A*). A putative plasmin cleavage site is present in mouse (i.e. AYK↓R$^{40}$, where the arrow indicates the cleavage site) and human (i.e. AYR↓R$^{44}$) OCN at the corresponding position (*Backes et al., 2000*). Bovine OCN, like the human protein, does not possess the *O*-glycosylation site found in mouse, and was shown to be cleaved by plasmin in vitro between R43 and R44 (*Novak et al., 1997*). Here we show that plasmin is sufficient to reduce the concentration non-glycosylated ucOCN, by more than 90% in two hours, confirming that mouse ucOCN is also a plasmin substrate. Supporting a role for plasmin in the cleavage of mouse OCN in vivo, OCN serum levels are decreased in mice deficient in plasmin activator inhibitor I (PAI-I), which have higher plasmin activity (*Tamura et al., 2013*). Interestingly, circulating plasmin activity increases with age in rats and humans (*Paczek et al., 2009*; *Paczek et al., 2008*), an observation which could in part explain the gradual reduction of serum OCN concentrations during aging (*Mera et al., 2016a*). Notably, a nonsynonymous rare variant (rs34702397) in OCN resulting in the conversion of R43 in a glutamine (Q) exists in humans of African ancestry and was nominally associated with insulin sensitivity index and glucose disposal in a small cohort of African Americans (*Das et al., 2010*). Since the R43Q variant eliminates the potential plasmin cleavage site in the C-terminal region, it will be interesting to further investigate if subjects carrying this variant have higher OCN circulating levels.

We also show that *O*-glycosylation protects mouse ucOCN from degradation in INS-1 cell cultures and consequently increases its biological activity. Interestingly, non-glycosylated ucOCN is stable when incubated in the INS-1 cell culture media at 37°C without cells (data not shown), suggesting that the proteolytic activity originates from the INS-1 cells and not from the culture media. The identity of a putative β-cell-derived protease responsible for ucOCN degradation remains unknown, since plasmin is expressed exclusively by the hepatocytes. Nevertheless, these results suggest that it may be more appropriate to use glycosylated ucOCN when studying the function of mouse OCN in cell culture.

We found that human OCN is not subjected to *O*-glycosylation and that consequently it has a reduced half-life in plasma ex vivo. The *O*-glycosylation sequence 'SVP$\underline{S}$PDP$^{11}$' of mouse OCN is conserved in the human protein, except for the amino acids corresponding to S5 and S8, which are replaced by a proline (P9) and a tyrosine (Y12) (i.e. 'PVP$\underline{Y}$PDP$^{15}$'). Remarkably, we could introduce

*O*-glycosylation into human OCN by a single amino acid change (Y12S). *O*-glycosylated human OCN is protected from degradation in plasma ex vivo similarly to the glycosylated mouse protein. Hence, this difference in the *O*-glycosylation status of OCN could potentially explain why circulating level of OCN in 1- to 6-month-old mice is 5–10 times higher than the level measured in young or adult human (*Table 1*).

It remains unknown if the increased half-life of *O*-glycosylated ucOCN will result in improved biological activity in vivo, although it was shown to be the case for other proteins (*Baudys et al., 1995*; *Runkel et al., 1998*; *Elliott et al., 2003*). Our cell-based assay does suggest that *O*-glycosylated ucOCN is active, at least on β-cells. We also showed that mouse OCN is endogenously *O*-glycosylated in vivo and that more than 80% of OCN, including the ucOCN fraction, is *O*-glycosylated in osteoblast supernatant and in serum. This suggests that the bioactive form of OCN is *O*-glycosylated in vivo in mice.

In summary, this work identified *O*-glycosylation as a previously unrecognized OCN PTM regulating its half-life in circulation in mice. This modification is not conserved in human, yet introducing *O*-glycosylation in human ucOCN also increases its half-life in plasma. These findings reveal an important difference between mouse and human OCN biology and also provide an approach to increase recombinant human OCN half-life in vivo, which might be relevant for the future development of OCN-based therapies for human diseases.

# Materials and methods

## Key resources table

| Reagent type (species) or resource | Designation | Source or reference | Identifiers | Additional information |
|---|---|---|---|---|
| Gene (*M. musculus*) | *Bglap* | GenBank | Gene ID: 12096 | Mouse osteocalcin gene 1 |
| Gene (*M. musculus*) | *Bglap2* | GenBank | Gene ID: 12097 | Mouse osteocalcin gene 2 |
| Gene (*Homo sapiens*) | *BGLAP* | GenBank | Gene ID: 632 | Human osteocalcin gene |
| Genetic reagent (*M. musculus*) | *Bglap-/-* | PMID:8684484 | *Bglap/Bglap2tm1Kry* RRID:MGI:3837364 | Genetic background: C57BL/6J |
| Genetic reagent (*M. musculus*) | *Furinfl/fl* | PMID:15471862 | *Furintm1Jwmc* RRID:MGI:3700793 | Genetic background: C57BL/6J |
| Genetic reagent (*M. musculus*) | BGLAP-Cre | PMID:12215457 | Tg(BGLAP-cre)1Clem RRID:IMSR_JAX:019509 | Genetic background: C57BL/6J |
| Genetic reagent (*M. musculus*) | C57BL/6J wildtype mice | The Jackson Laboratory | Stock No: 000664 RRID:IMSR_JAX:000664 | For primary osteoblasts preparation |
| Cell line (*R. norvegicus*) | INS-1 832/3 | Millipore-Sigma | SCC208 RRID:CVCL_ZL55 | |
| Cell line (*C. griseus*) | Chinese hamster ovary (CHO-K1) cells | ATCC | CCL-61 RRID:CVCL_0214 | |
| Cell line (*C. griseus*) | Chinese hamster ovary ldlD cells (CHO-ldlD) | PMID:3948246 | RRID:CVCL_1V03 | Cell maintained in N. Seidah lab. |
| Cell line (*M. musculus*) | Primary osteoblasts | This paper | | Prepared from C57BL/6J wildtype mice newborn calvaria |
| Cell line (*H. sapiens*) | Human embryonic kidney cells HEK293 | ATCC | CRL-1573 RRID:CVCL_0045 | |
| Cell line (*H. sapiens*) | COSMC knockout HEK293 cells (*C1GALT1C1-/-*) | PMID:23584533 | RRID:CVCL_S025 | Cell maintained in H. Clausen lab. |
| Cell line (*H. sapiens*) | GALNT3/6 knockout HEK293 cells | PMID:31040225 | | Cell maintained in H. Clausen lab. |
| Cell line (*H. sapiens*) | GALNT3 knockout HEK293 cells | PMID:31040225 | | Cell maintained in H. Clausen lab. |
| Cell line (*H. sapiens*) | GALNT6 knockout HEK293 cells | PMID:31040225 | | Cell maintained in H. Clausen lab. |

*Continued on next page*

*Continued*

| Reagent type (species) or resource | Designation | Source or reference | Identifiers | Additional information |
|---|---|---|---|---|
| Cell line (*H. sapiens*) | GALNT1/2/3 knockout HEK293 cells | PMID:31040225 | | Cell maintained in H. Clausen lab. |
| Transfected construct (*M. musculus*) | pIRES2-EGFP-mOCN-V5 | This paper | | To express mouse OCN V5 tagged in primary osteoblasts, CHO-K1, CHO-ldlD and HEK293 |
| Transfected construct (*M. musculus*) | pIRES2- EGFP-mOCN (S5A/S8/AT15A) -V5 | This paper | | To express (S5A/S8/AT15A) mutant mouse OCN V5 tagged in primary osteoblasts |
| Transfected construct (*M. musculus*) | pIRES2- EGFP-mOCN (S29A/T36A/T45A) -V5 | This paper | | To express (S29A/T36A/T45A) mutant mouse OCN V5 tagged in primary osteoblasts |
| Transfected construct (*M. musculus*) | pIRES2- EGFP-mOCN (6XST→6XA)-V5 | This paper | | To express (6XST→6XA) mutant mouse OCN V5 tagged in primary osteoblasts |
| Transfected construct (*M. musculus*) | pIRES2- EGFP-mOCN (S5A)-V5 | This paper | | To express (S5A) mutant mouse OCN V5 tagged in primary osteoblasts |
| Transfected construct (*M. musculus*) | pIRES2- EGFP-mOCN (S8A)-V5 | This paper | | To express (S8A) mutant mouse OCN V5 tagged in primary osteoblasts |
| Transfected construct (*M. musculus*) | pIRES2- EGFP-mOCN (T15A)-V5 | This paper | | To express (T15A) mutant mouse OCN V5 tagged in primary osteoblasts |
| Transfected construct (*H. sapiens*) | pIRES2-EGFP-hOCN-V5 | This paper | | To express human OCN V5 tagged in primary osteoblasts |
| Transfected construct (*H. sapiens*) | pIRES2-EGFP-hOCN (Y12S)-V5 | This paper | | To express (Y12S) mutant human OCN V5 tagged in primary osteoblasts |
| Transfected construct (*H. sapiens*) | pIRES2-EGFP-hOCN (Y12L)-V5 | This paper | | To express (Y12L) human OCN V5 tagged in primary osteoblasts |
| Transfected construct (*H. sapiens*) | pcDNA3.1-Fc-hinge-Thr-mOCN | This paper | | Used to produce *O*-gly ucOCN in HEK293 |
| Transfected construct (*H. sapiens*) | pcDNA3.1-Fc-hinge-Thr-hOCN (Y12S) | This paper | | Used to produce *O*-gly uc-hOCN in HEK293 |
| Antibody | Anti-GFP, mouse monoclonal, clones 7.1 and 13.1 | Sigma-Aldrich | 11814460001 RRID:AB_390913 | WB (1:1000) |
| Antibody | Anti-V5, mouse monoclonal, clone V5-10 | Sigma-Aldrich | V8012 RRID:AB_261888 | WB (1:3000) |
| Antibody | Anti–β-actin, mouse monoclonal, clone AC-15 | Sigma-Aldrich | A5441 RRID:AB_476744 | WB (1:7000) |
| Antibody | Anti-Gla-OCN goat polyclonal antibody (recognize amino acids 11–26 of carboxylated mature mouse OCN) | PMID:20570657 | | WB (1:3000) ELISA (2 µg/ml) |
| Antibody | Anti-CTERM OCN goat polyclonal antibody recognize amino acids 26–46 of mature mouse OCN | PMID:20570657 | | WB (1:3000) ELISA (1:600) IP (1:100) |
| Antibody | Anti-MID OCN goat polyclonal antibody recognize amino acids 11 to 26 of mature mouse OCN | PMID:20570657 | | ELISA (1.5 µg/ml) |
| Recombinant DNA reagent | pTT5-Fc1_CTL | PMID:23951290 | | Used as PCR template to amplify Fc and hinge region |

*Continued on next page*

*Continued*

| Reagent type (species) or resource | Designation | Source or reference | Identifiers | Additional information |
|---|---|---|---|---|
| Peptide, recombinant protein | Collagenase type 2 | Worthington Biochemical Corporation | LS004176 | For primary osteoblasts preparation |
| Peptide, recombinant protein | O-Glycosidase and Neuraminidase Bundle | NEB | E0540S | Deglycosylation assay |
| Peptide, recombinant protein | Thrombin | GE Healthcare Life Sciences | 27-0846-01 | Protein purification |
| Peptide, recombinant protein | Human plasmin | Sigma | P1867 | |
| Chemical compound, drug | Warfarin | Santa Cruz Biotechnology | sc-205888 | VKORC1 inhibitor |
| Chemical compound, drug | Decamoyl-RVKR-CMK | Tocris | 3501/1 | Furin inhibitor |
| Chemical compound, drug | N-acetylgalacto saminyltransferase inhibitor (GalNAc-bn) | Sigma | 200100 | GalNAc-Ts inhibitor |
| Chemical compound, drug | Benzamidine sepharose | GE healthcare | 17-5123-10 | Protein purification |
| Chemical compound, drug | Pepstatin A | Sigma | P5318 | Aspartyl proteases inhibitor |
| Chemical compound, drug | Talabostat | Tocris, | 3719/10 | FAP inhibitor |
| Chemical compound, drug | Phenylmethylsulfonyl fluoride (PMSF) | Amresco | 329-98-6 | Serine proteases inhibitor |
| Chemical compound, drug | Vitamin $K_1$ | Sigma | V3501 | Cofactor for gamma carboxylation |
| Commercial assay or kit | HiTrap protein A high performance | GE Healthcare Life Sciences | GE29-0485-76 | Protein purification |
| Commercial assay or kit | Human ucOCN ELISA | BioLegend (PMID:31935114) | 446707 | |
| Commercial assay, kit | JetPrime | Polypus transfection | 114–15 | |
| Commercial assay, kit | Lipofectamine 2000 | Thermo Fisher | 11668019 | |
| Software, algorithm | Prism version 7.03 | GraphPad | RRID:SCR_002798 | |
| Software, algorithm | Xcalibur 4.0 | Thermo Fisher Scientific | RRID:SCR_014593 | |

## Animal models

The *Furin*[fl/fl];BGLAP-Cre mice were generated by breeding *Furin*[fl/fl] (*Furin*[tm1Jwmc]/*Furin*[tm1Jwmc]) mice with BGLAP-Cre (Tg(BGLAP-cre)1Clem) transgenic mice that express Cre recombinase under the control of human OCN promoter as described previously (*Al Rifai et al., 2017*). *Bglap-/-* mice (*Bglap/Bglap2*[tm1Kry]/*Bglap/Bglap2*[tm1Kry]) were generated using homologous recombination to replace OCN1 (*Bglap1*) and OCN2 (*Bglap2*) genes in the mouse *Bglap* cluster with a neomycin resistance cassette (*Ducy et al., 1996*). All strains used in this study were backcrossed on a C57BL/6J genetic background more than 10 times and maintained under 12 hr dark/12 hr light cycles in a specific pathogen–free (SPF) animal facility at IRCM. Male mice were used in all experiments, and they were fed a normal chow diet. All animal use complied with the guidelines of the Canadian Committee for Animal Protection and was approved by IRCM Animal Care Committee.

## DNA constructs

Mouse pro-OCN cDNA was cloned into the pIRES2-EGFP-V5 plasmid in EcoRI and AgeI cloning sites. S5A/S8/AT15A pro-OCN, S29A/T36A/T45A pro-OCN and S5S/S8A/T15A/S29A/T36A/T45A

(i.e. 6XST→6XA) pro-OCN mutant were purchased from Thermo Fisher. Human pre-pro-OCN cDNA cloned into pcDNA3 was purchased from GenScript. Each construct was used as PCR template for amplification and to introduce EcoRI and AgeI cloning sites and cloned in pIRES2-EGFP-V5 plasmid. Point mutations in mouse pro-OCN (S5A, S8A, T15A) and Y12S in human pro-OCN were generated by site-directed mutagenesis using specific primer (*Appendix 1—table 1*).

The cDNA coding of the Fc and hinge region of human immunoglobulin flanked with HindIII-BamHI restriction sites was amplified using standard PCR (*Appendix 1—table 1*) and pTT5-Fc1_CTL vector as a template (*Saavedra et al., 2013*). The PCR product was cloned in pcDNA3.1-myc-His B in HindIII-BamHI cloning site, generating the pcDNA3.1-Fc-hinge-myc-His vector. The cDNA coding for mature hOCN$^{(Y12S)}$ was generated using pIRES2-EGFP-hOCN (Y12S)-V5 as a template, to which a thrombin (Thr) cleavage site was added at the N-terminus and BglII and EcoRI restriction sites were introduced by standard PCR amplifications. The Thr-hOCN$^{(Y12S)}$ product was cloned in the pcDNA3.1-Fc-hinge-myc-His vector. The generated vector pcDNA3.1-Fc-hinge-Thr-hOCN$^{(Y12S)}$ is an expression vector of human OCN fusion protein composed of the Fc and hinge region of human IgG1, thrombin cleavage site and human OCN (Y12S). Mouse OCN fused to Fc were generated following the same procedure and using different primers (*Appendix 1—table 1*).

## Cell culture and transfection

All cell lines tested negative for mycoplasma. The identity of the cell lines was confirmed by STR when obtained from ATCC or Millipore. Otherwise, the identity was based on the phenotype reported by the providing investigator (e.g. lack of glycosylation for CHO-ldld, lack of expression of the GALNTs for the knockout HEK293 cell lines).

Primary osteoblasts were prepared following a previously described protocol (*Ferron et al., 2015*). In brief, calvariae were collected from 3 days old mice and washed with 1 × PBS and digested 2 times for 10 min in digestion solution (αMEM, 0.1 mg/ml collagenase type 2 [Worthington Biochemical Corporation] and 0.00075% trypsin) that was discarded after incubation. Following two 30 min incubations, the digestion solutions were collected, centrifuged and cells recovered were cultured in αMEM supplemented with 10% FBS, penicillin and streptomycin (PS), and L-glutamine. Culture media was supplemented with 5 mM β-glycerophosphate and 100 µg/ml L-ascorbic acid to induce osteoblasts differentiation and it was replaced every 2 days for 21 days.

Primary osteoblasts were transfected using jetPRIME Reagent (Polypus transfection). After an overnight incubation, media were changed to secretion media (FBS-free αMEM plus 2 mM L-glutamine, PS). After 24 hr of secretion, media were collected, and cells were lysed in protein lysis buffer (20 mM Tris-HC pH 7.4, 150 mM NaCl, 1 mM EDTA, 1 mM EGTA, 1% Triton, 1 mM PMSF, and 1 × protease inhibitor cocktail) and analyzed by western blotting. In some experiments, osteoblasts were treated with the γ-carboxylation inhibitor warfarin (50 µM; Santa Cruz Biotechnology), the *N*-acetylgalactosaminyltransferase inhibitor GalNAc-bn (2 mM, Sigma) or the proprotein convertase inhibitor Dec-RVKR-CMK (50 µM, Tocris), combined with 22 µM vitamin K$_1$ (Sigma).

Chinese hamster ovary (CHO) cells, originally purchased from ATCC, and Chinese hamster ovary ldlD cells (CHO-ldlD; originating from the M. Krieger laboratory *Kingsley et al., 1986*) were cultured in DMEM-F12 containing PS and 5% FBS for CHO cells or 3% FBS for CHO-ldlD cells and transfected using Lipofectamine 2000 (Thermo Fisher) following standard protocol. Secretion was performed in DMEM-F12 media supplemented with PS and 22 µM VK$_1$. In some experiments, CHO-ldlD culture, transfection and secretion media was supplemented with 0.1 mM galactose and/or 1 mM *N*-acetyl-galactosamine (GalNAc) to rescue the *O*-glycosylation defect as previously reported (*Kingsley et al., 1986*).

Human embryonic kidney cells HEK293 were originally purchased from ATCC. *C1GALT1C1 knockout HEK293sc (HEK293 simple cell or COSMC) cells* and *GALNTs* deficient HEK293 cells were generated using Zinc-finger nuclease (ZFN) gene editing as described previously (*Goth et al., 2015*; *Schjoldager et al., 2012*; *Steentoft et al., 2013*; *Steentoft et al., 2011*; *Goth et al., 2017*). Cells were transfected using Lipofectamine 2000 reagent and secretion was performed over 24 hr in EMEM supplemented with PS and 22 µM VK1. In some experiments, HEK293 cells were treated with warfarin, GalNAc-bn or Dec-RVKR-CMK combined with 22 µM vitamin K$_1$.

For western blot analysis, proteins were resolved on 15% Tris-tricine gel and blotted overnight with indicated antibody. Antibody used in this study are: anti-V5 (mouse, clone V5-10, V8012; Sigma-Aldrich), anti–β-actin (mouse, clone AC-15, A5441; Sigma-Aldrich), anti-GFP (mouse, clones

7.1 and 13.1, 11814460001; Sigma), anti-Gla-OCN goat antibody which recognize amino acids 11–26 of carboxylated mature OCN and anti-Cterm OCN goat antibody which recognize amino acids 26–46 of mature mouse OCN (*Ferron et al., 2010b*).

### In-vitro de-glycosylation assay

Flushed mouse femur and tibia from C57BL/6J were homogenized in lysis buffer (20 mM Tris-HCl pH 7.4, 150 mM NaCl, 1 mM EDTA, 1 mM EGTA, 1% Triton, 1 mM PMSF, and 1 × protease inhibitors cocktail). Tissue homogenates were then centrifuged for 10 min at 4000 rpm to remove insoluble material. In-vitro de-glycosylation assay was performed on 10 µg of bone homogenate. Briefly, proteins were denatured in denaturing buffer (0.5% SDS, 40 mM DTT) at 95°C for 5 min and incubated with 80000 units of $O$-glycosidase and 100 units of neuraminidases for 4 hr at 37 °C following the NEB kit protocol (E0540S; NEB). Samples were resolved on 15% Tris-tricine SDS-PAGE gel and blotted using anti-Cterm OCN goat antibody.

### Top-down LC-MS/MS analysis MS analysis of OCN in osteoblasts secretion media and bone extracts

Differentiated osteoblast secretion medium was spun down at 1500 rpm for 5 min to remove cells debris. The supernatant was then incubated overnight at 4°C with anti-Cterm OCN antibody in the presence of 1 × protease inhibitors cocktail. For bone extract, flushed femur and tibia from wild type mice were homogenized in lysis buffer containing (20 mM Tris-HCl pH 7.4, 150 mM NaCl, 1 mM EDTA, 1 mM EGTA, 1% Triton, 1 mM PMSF, and 1 × protease inhibitors cocktail). 100 µg of protein homogenate was diluted in 1,6 ml of 100 mM phosphate buffer pH 7.4 and incubated overnight at 4°C with anti-OCN antibody. After overnight incubation, samples were centrifuged at 10000 rpm for 10 min to remove precipitate and supernatant was incubated with protein-G agarose beads pre-washed with 1X PBS. After for 4 hr of rotation at 4°C, beads were spun down, washed twice with 1X PBS and three times with 50 mM Ammonium Bicarbonate pH 8.0. OCN was then eluted with 100 µl of 0.5 M NH$_4$OH, snap frozen in liquid nitrogen and evaporated under vacuum using speedvac concentrator (Thermo scientific).

Samples were diluted in 25% ACN 0.3%TFA and loaded onto a 50 × 4.6 mm PLRP-S 300A column (Agilent Technologies) connected to an Accela pump (Thermo Scientific) and a RTC autosampler (Pal systems). The buffers used for chromatography were 0.1% formic acid (buffer A) and 100% acetonitrile/0.1% formic acid (buffer B). Proteins and peptides were eluted with a two slopes gradient at a flowrate of 120 µL/min. Solvent B first increased from 12% to 50% in 4.5 min and then from 50% to 70% in 1.5 min. The HPLC system was coupled to a Q Exactive mass spectrometer or an Orbitrap Fusion (Thermo Scientific) through an Ion Max electrospray Ion Source equipped with a HESI-II probe and operated in positive ion mode. The spray and S-lens voltages were set to 3.6 kV and 60 V, respectively. Capillary temperature was set to 225°C. Full scan MS survey spectra (m/z 600–2000) in profile mode were acquired in the Orbitrap with a resolution of 70,000 or 120,000 with a target value at 3e6. The four most intense protein/peptide ions were fragmented in the HCD (higher-energy collision dissociation) collision cell upon collision with nitrogen gas and analyzed in the Orbitrap with a target value at 5e5 and a normalized collision energy at 33. The data acquisition software were Xcalibur 3.1 and Q Exactive 2.8 SP1 for the Q Exactive instrument and Xcalibur 4.0 and Tune 2.0 for the Orbitrap Fusion instrument. Data processing protocol: the identification of the different forms of OCN was performed by manual denovo sequencing using Qual Browser (Xcalibur 4.0). The source files for the proteomics analyses can be downloaded through this link: https://doi.org/10.6084/m9.figshare.13259891.v1.

### *Galnts* expression in osteoblasts

RNA was extracted from non-differentiated and differentiated calvariae osteoblasts using Trizol reagent (Thermo Fisher Scientific) following standard protocol. RNA was treated with DNAse I and reverse transcribed using poly dT primers, random primers and MMLV reverse transcriptase (Thermo Fisher Scientific). QPCR was performed on standards of diluted genomic DNA and cDNA products using specific primers (*Appendix 1—table 1*) on a ViiA 7 Real-Time PCR system (Thermo Fisher Scientific). *Galnts* gene copy numbers were calculated using the genomic DNA as a standard curve and variation between biological replicate was normalized using *Actb* expression level.

## Generation of stable HEK293 clones expressing mouse and human OCN fused to the Fc region of human immunoglobulin

To generate stable clonal cell lines expressing glycosylated human and mouse OCN, HEK293 were transfected with pcDNA3.1-Fc-hinge-Thr-hOCN$^{Y12S}$ and pcDNA3.1-Fc-hinge-Thr-OCN respectively using Lipofectamine 2000 reagent. Following 48 hr of transfection, cells were trypsinized and resuspended in sorting buffer containing (1 × sterile PBS, 2% FBS and 1 mM EDTA). Cells were sorted at a concentration of 5–10 cells/well in 96 well plates containing the selection media EMEM, 10% FBS supplemented with G418 sulfate (500 µg/ml; Wisent). Following two weeks of selection, isolated colonies appeared and the expression of mouse and human OCN was assessed using ELISA assay described below. Clones expressing the highest levels of OCN were amplified and frozen.

## Purification of mouse and human OCN fused to the Fc region of human immunoglobulin

TM102F12 clone expressing IgFc-mOCN fusion protein and 22H5 clone expressing IgFc-hOCN$^{Y12S}$ fusion protein were cultured in triple layer 175 cm$^2$ flasks. After reaching 100% confluency, cells were kept in secretion media (EMEM media supplemented with 1% FBS and 10 µM warfarin to block γ-carboxylation) for 72 hr. Secretion media was collected, filtered with 0.45 µm filter, and media was pH buffered with 10 × binding buffer (0.2 M phosphate buffer, pH 7). This cell supernatant was then loaded into protein A affinity column (HiTrap protein A high performance, GE29-0485-76; GE Healthcare Life Sciences,) using liquid chromatography system (GE AKTA Prime Plus). Column was then washed with 20 ml of 1 × binding buffer (0.02 M phosphate buffer, pH 7) and 5 ml of filtered 1 × PBS. To release OCN from the column, OCN fusion protein was digested with thrombin (27-0846-01, GE Healthcare Life Sciences) and eluted with 1 × PBS. Thrombin was subsequently removed using benzamidine sepharose (17-5123-10, GE healthcare). Mouse and human OCN purity were assessed using Coomassie staining and liquid chromatography-mass spectrometry (LC-MS) analysis compared to purified non-glycosylated mouse or human ucOCN. Mouse OCN was quantified using ELISA assay as described previously (*Ferron et al., 2010b*). Human ucOCN measurements were performed using a commercially available human ucOCN ELISA (BioLegend, 446707) (*Lacombe et al., 2020*), which recognizes equally glycosylated and non-glycosylated human OCN (*Figure 5—figure supplement 2*). The capture antibody in this assay is a mouse monoclonal antibody (8H4) specific to the C-terminal region of human OCN (i.e. amino acids 30 to 49). The detection antibody is a mouse monoclonal antibody (4B6) specific to the mid-region of human ucOCN (i.e. amino acids 12 to 28 in ucOCN).

## Ex vivo half-life and plasmin enzymatic assays

The mouse OCN ex vivo half-life assays were performed with fresh plasma (lithium heparin) collected from four independent *Bglap-/-* mice. Glycosylated OCN produced in HEK293 cells and non-glycosylated OCN, produced in bacteria as previously described (*Ferron et al., 2012*), were incubated at 100 ng/ml in plasma at 37°C and OCN level was measured at indicated time points using the total mouse OCN ELISA assay described previously (*Ferron et al., 2010b*). In this assay, the capture antibody is a goat polyclonal antibody directed against the central part (amino acids 11 to 26) of mouse OCN (anti-MID OCN) and recognizing with equal affinity the uncarboxylated and carboxylated OCN proteins. The detection antibody is a goat polyclonal antibody directed against the C-terminal region (amino acids 26 to 46) of mouse OCN (anti-CTERM OCN). This assay detects with the same sensitivity glycosylated and non-glycosylated mouse OCN at concentration of 50 ng/ml or less (*Figure 3—figure supplement 2*). Therefore, all samples were diluted to be in this range before being measured by the ELISA. Human OCN half-life assay was performed ex vivo using *Bglap-/-* mice plasma and human OCN at 60 ng/ml. Human OCN level was measured at different time point using the human ucOCN ELISA described above. At the experiment start point (T0), an aliquot of plasma was diluted in ELISA assay buffer and kept on ice for OCN measurements. In some experiments, plasma was heat-inactivated for 30 min at 56°C, or treated with EDTA (10 mM, Wisent), phenylmethylsulfonyl fluoride (PMSF, 10 mM, Sigma), pepstatin A (Pep A, 10 µM, Sigma) which inhibits aspartic proteases (pepsin, cathepsin D, renin, chymosin), RVKR (Dec-RVKR-CMK, 50 µM; Tocris) or talabostat (10 mM; Tocris). Results are calculated in percentage relative to the initial concentration of non-glycosylated and glycosylated OCN respectively.

The plasmin enzymatic assay was performed on plasma ex vivo in the presence of non-glycosylated OCN and glycosylated OCN as follows: plasma from *Bglap-/-* mice was heat-inactivated for 30 min at 56°C, then diluted two times in the enzyme assay buffer containing 100 mM Tris buffer pH 7.5. Non-glycosylated OCN or glycosylated OCN was incubated at 50 ng/ml in the diluted plasma for two hours in the presence of different concentration of human plasmin (Sigma) ranging from 0 to 0.3 U/ml. After two hours, OCN level was measured using total mouse OCN ELISA. Results are represented as percentage of initial concentration of non-glycosylated and glycosylated OCN respectively.

### In vivo half-life assay

For in vivo half-life assay, *Bglap-/-* male mice were injected intraperitoneally with 40 or 80 ng/g of body weight of mouse *O*-glycosylated ucOCN or non-glycosylated ucOCN, serum OCN level was analyzed at indicated time points using total mouse OCN ELISA. In all ex vivo and in vivo studies, mouse or human proteins were prepared in saline solution (0.9% NaCl) containing BSA (35 mg/ml) as a carrier.

### INS-1 832/3 treatment with ucOCN

The INS-1 832/3 rat insulinoma cell line (Millipore-Sigma) was cultured in RPMI-1640 with 11.1 mM D-glucose, supplemented with 2 mM L-glutamine, 1 mM sodium pyruvate, 10 mM HEPES, 0.05 mM β-mercaptoethanol and 10% FBS (*Hohmeier et al., 2000*). On day 1, cells were plated at $3 \times 10^5$ cells per well in 12-well plates and on day 5, cells were washed with PBS and incubated with culture media containing 5 mM D-glucose. Twenty-four hours later, cells were washed with PBS and incubated for 4 hr in culture media containing 1% FBS, 5 mM D-glucose and 0.1% BSA. OCN was added for 8 hr before supernatants and cells were harvested. ucOCN and *O*-gly ucOCN concentration were measured in supernatants by ELISA and cells were lyzed and RNA extracted as described in the material and methods section. Gene expression was analyzed using rat specific primers for *Ins1* and *Gapdh*.

### Statistics

Statistical analyses were performed using GraphPad Prism software version 7.03. Results are shown as the mean ± SEM. For single measurements, an unpaired, 2-tailed Student's *t* test was used, while one-way or two-way ANOVA followed by Bonferroni's post-test were used for comparison of more than two groups. For repeated measurements (e.g. half-life study ex vivo and in vivo), a repeated-measurement two-way ANOVA followed by Bonferroni's post-test was used. A *P* value of less than 0.05 was considered statistically significant. All experiments were repeated at least three times or performed on at least three independent animals or included at least three independent biological replicates. In the in vivo half-life experiments, the *Bglap-/-* mice were randomized before the experiment into experimental groups such that the average body weight of each group was similar.

## Acknowledgements

We thank Dr. Nabil Seidah for providing reagents as well as the CHO-ldld cells. We thank Dr. John Creemers for providing the *Furin* floxed mice. This work was supported by funding from the Canada Research Chair program (MF), the Canadian Institutes of Health Research (MF, MOP-133652 and PJT-159534) and the Natural Sciences and Engineering Research Council of Canada (RGPIN-2016–05213, to MF), and the Danish National Research Foundation (DNRF107, to HC). OAR received scholarships from IRCM and FRQS.

## Additional information

### Funding

| Funder | Grant reference number | Author |
| --- | --- | --- |
| Canadian Institutes of Health Research | Operation fund MOP-133652 | Mathieu Ferron |

| Canadian Institutes of Health Research | Project Operating fund PJT-159534 | Mathieu Ferron |
|---|---|---|
| Natural Sciences and Engineering Research Council of Canada | Discovery grant RGPIN-2016-05213 | Mathieu Ferron |
| Danmarks Grundforsknings-fond | DNRF107 | Henrik Clausen |
| Fonds de Recherche du Québec - Santé | Doctoral scholarship | Omar Al Rifai |
| Institut de Recherche Clinique De Montréal | Doctoral scholarship | Omar Al Rifai |
| Canada Research Chairs | | Mathieu Ferron |

The funders had no role in study design, data collection and interpretation, or the decision to submit the work for publication.

### Author contributions

Omar Al Rifai, Conceptualization, Formal analysis, Validation, Investigation, Visualization, Methodology, Writing - original draft; Catherine Julien, Investigation, Methodology, Writing - review and editing; Julie Lacombe, Conceptualization, Formal analysis, Supervision, Validation, Investigation, Visualization, Methodology, Writing - review and editing; Denis Faubert, Resources, Formal analysis, Validation, Investigation, Visualization, Methodology, Writing - review and editing; Erandi Lira-Navarrete, Investigation; Yoshiki Narimatsu, Investigation, Visualization, Writing - review and editing; Henrik Clausen, Resources, Supervision, Writing - review and editing; Mathieu Ferron, Conceptualization, Resources, Formal analysis, Supervision, Funding acquisition, Writing - original draft, Project administration, Writing - review and editing

### Author ORCIDs

Mathieu Ferron (ID) https://orcid.org/0000-0002-5858-2686

### Ethics

Animal experimentation: All animal use complied with the guidelines of the Canadian Committee for Animal Protection and was approved by IRCM Animal Care Committee (protocol # 2016-14 MF).

### Decision letter and Author response

Decision letter https://doi.org/10.7554/eLife.61174.sa1
Author response https://doi.org/10.7554/eLife.61174.sa2

## Additional files

### Supplementary files

- Transparent reporting form
- Reporting standard 1. Check list for the "Reporting guidelines for mass spectrometry".

### Data availability

All the numerical data and the original western blots are available in the source data Excel file submitted with the manuscript. The raw proteomics data have been uploaded to a public server.

The following dataset was generated:

| Author(s) | Year | Dataset title | Dataset URL | Database and Identifier |
|---|---|---|---|---|
| Rifai OA, Julien C, Lacombe J, Faubert D, Lira-Navarrete E, Narimatsu YJL, | 2020 | MS raw files for The half-life of the bone-derived hormone osteocalcin is regulated through O-glycosylation in mice, but not in | https://doi.org/10.6084/m9.figshare.13259891.v1 | figshare, 10.6084/m9.figshare.13259891.v1 |

Clausen H, Ferron M                    humans

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

# Appendix 1

**Appendix 1—table 1.** Additional key resources.

| Reagent type (species) or resource | Designation | Source or reference | Identifiers | Additional information |
|---|---|---|---|---|
| Sequence-based reagent | mOCN-For-EcoRI | This paper | PCR primers (Cloning of mOCN in pIRES2-EGFP-V5) | AATTGAATTCgCcaccatgaggaccctctctc |
| Sequence-based reagent | mOCN-For-EcoRI | This paper | PCR primer (cloning of mOCN in pIRES2-EGFP-V5) | AATTGAATTCGCCACCATGAGGACCCTCTCTC |
| Sequence-based reagent | mOCN-Rev-Stop-AgeI | This paper | PCR primer (cloning of mOCN in pIRES2-EGFP-V5, No V5 tagged protein) | AATTACCGGTCTAAATAGTGATACCGTAGATG |
| Sequence-based reagent | mOCN-Rev-AgeI | This paper | PCR primer (cloning of mOCN in pIRES2-EGFP-V5) | AATTACCGGTAATAGTGATACCGTAGATGCG |
| Sequence-based reagent | mOCNSTT-stop-Age1-Rev | This paper | PCR primer (cloning of S29A/T36A/T45A mOCN in pIRES2-EGFP-V5, No V5 tagged protein) | AATTACCGGTCTAAATAGCGATACCGTAGATG |
| Sequence-based reagent | mOCNSTT-Age1-Rev | This paper | PCR primer (cloning of S29A/T36A/T45A mOCN in pIRES2-EGFP-V5) | AATTACCGGTAATAGCGATACCGTAGATGCG |
| Sequence-based reagent | mOCN-S5A-For | This paper | PCR primer (mutagenesis of Serine five to Alanine in mOCN) | TACCTTGGAGCCGCCGTCCCCAGCCCA |
| Sequence-based reagent | mOCN-S5A-Rev | This paper | PCR primer (mutagenesis of Serine five to Alanine in mOCN) | TGGGCTGGGGACGGCGGCTCCAAGGTA |
| Sequence-based reagent | mOCN-S8A-For | This paper | PCR primer (mutagenesis of Serine eight to Alanine in mOCN) | GCCTCAGTCCCCGCCCCAGATCCCCTG |
| Sequence-based reagent | mOCN-S8A-Rev | This paper | PCR primer (mutagenesis of Serine eight to Alanine in mOCN) | CAGGGGATCTGGGGCGGGGACTGAGGC |
| Sequence-based reagent | mOCN-T15A-For | This paper | PCR primer (mutagenesis of Threonine 15 to Alanine in mOCN) | CTGGAGCCCGCCCGGGAGCAG |
| Sequence-based reagent | mOCN-T15A-Rev | This paper | PCR primer (mutagenesis of Threonine 15 to Alanine in mOCN) | CTGCTCCCGGGC GGGCTCCAG |
| Sequence-based reagent | hOCN-EcoRI-For | This paper | PCR primer (cloning human osteocalcin in pIRES2-EGFP-V5) | AATTGAATTCGCCACCATGAGAGCCCTCACACTCCT |

*Continued on next page*

*Appendix 1—table 1 continued*

| Reagent type (species) or resource | Designation | Source or reference | Identifiers | Additional information |
|---|---|---|---|---|
| Sequence-based reagent | hOCN-AgeI-Rev | This paper | PCR primer (cloning human osteocalcin in pIRES2-EGFP-V5) | AATT ACCGGT GACCGGGCCGTAGAAGCG |
| Sequence-based reagent | hOCN-Y12S-For | This paper | PCR primer (mutagenesis of Tyrosine 12 to Serine in hOCN) | GCCCCAGTCCCCAGCCCGGATCCCCTG |
| Sequence-based reagent | hOCN-Y12S-Rev | This paper | PCR primer (mutagenesis of Tyrosine 12 to Serine in hOCN) | CAGGGGATCCGGGCTGGGGACTGGGGC |
| Sequence-based reagent | hOCN-Y12L-For | This paper | PCR primer (mutagenesis of Tyrosine 12 to Leucine in hOCN) | GCCCCAGTCCCCCTACCGGATCCCCTG |
| Sequence-based reagent | hOCN-Y12L-Rev | This paper | PCR primer (mutagenesis of Tyrosine 12 to Leucine in hOCN) | CAGGGGATCCGGTAGGGGGACTGGGGC |
| Sequence-based reagent | HindIII-FchIgG1 -For | This paper | PCR primer (amplification of FC fragment+ hinge region in pTT5FC-CTL plasmid, and cloning in pcDNA3 in HindIII-BamHI) | AATTAAGCTTGCCACCATGGAGTTTGGGCTG |
| Sequence-based reagent | BamHI-FchIgG1-Rev | This paper | PCR primer (amplification of FC fragment+ hinge region in pTT5FC-CTL plasmid, and cloning in pcDNA3 in HindIII-BamHI) | AATTGGATCCTGGGCACGGTGGGCATGTG |
| Sequence-based reagent | BamHI-Thrombin-mOCN-For | This paper | PCR primer (cloning of Thrombin mOCN in pcDNA3 FchIgG1 using BamHI-EcoRI) | AATTGGATCCCTGGTTCCGCGTGG ATCTTACCTTGGAGCCTCAGTCC |
| Sequence-based reagent | EcoRI-mOCN-Rev | This paper | PCR primer (cloning of Thrombin mOCN in pcDNA3 FchIgG1 using BamHI-EcoRI) | AATTGAATTCCTAAATAGTGATACCGTAGATG |
| Sequence-based reagent | BglII-Thrombin-hOCN- For | This paper | PCR primer (cloning of Thrombin hOCN (Y12S) in pcDNA3 FchIgG1 using BglII-EcoRI) | AATTAGATCTCTGGTTCCGCGTGGA TCTTACCTGTATCAATGGCTGG |
| Sequence-based reagent | EcoRI-hOCN-Rev | This paper | PCR primer (cloning of Thrombin hOCN (Y12S) in pcDNA3 FchIgG1 using BglII-EcoRI) | AATTGAATTCCTAGACCGGGCCGTAGAAGCGC |
| Sequence-based reagent | GalnT1-For | This paper | QPCR primer (amplify *Galnt1, M. musculus*) | GCAGCATGTGAACAGCAATCA |

*Continued on next page*

*Appendix 1—table 1 continued*

| Reagent type (species) or resource | Designation | Source or reference | Identifiers | Additional information |
|---|---|---|---|---|
| Sequence-based reagent | GalnT1-Rev | This paper | QPCR primer (amplify *Galnt1, M. musculus*) | GCTGAGGTAGCCCAGTCAATC |
| Sequence-based reagent | GalnT2-For | This paper | QPCR primer (amplify *Galnt2, M. musculus*) | GGCAACTCCAAACTGCGACA |
| Sequence-based reagent | GalnT2-Rev | This paper | QPCR primer (amplify *Galnt2, M. musculus*) | TCAACAAACTGGGCCGGTG |
| Sequence-based reagent | GalnT3-For | This paper | QPCR primer (amplify *Galnt3, M. musculus*) | ACTTAGTGCCATGTGACGCA |
| Sequence-based reagent | GalnT3-Rev | This paper | QPCR primer (amplify *Galnt3, M. musculus*) | GGGTTTCTGCAGCGGTTCTA |
| Sequence-based reagent | GalnT4-For | This paper | QPCR primer (amplify *Galnt4, M. musculus*) | CAAAACTGCCCCAAAGACGG |
| Sequence-based reagent | GalnT4-Rev | This paper | QPCR primer (amplify *Galnt4, M. musculus*) | CGCTCTGCTGCTAGCCTATT |
| Sequence-based reagent | GalnT5-For | This paper | QPCR primer (amplify *Galnt5, M. musculus*) | CCCTGAAACTGGCTGCTTGT |
| Sequence-based reagent | GalnT5-Rev | This paper | QPCR primer (amplify *Galnt5, M. musculus*) | ATGGAGAGAAATTCAGTCAGCAA |
| Sequence-based reagent | GalnT6-For | This paper | QPCR primer (amplify *Galnt6, M. musculus*) | CCAGCTCTGGCTGTTTGTCTA |
| Sequence-based reagent | GalnT6-Rev | This paper | QPCR primer (amplify *Galnt6, M. musculus*) | TTGGGCCAAGTAGCATGTGA |
| Sequence-based reagent | GalnT7-For | This paper | QPCR primer (amplify *Galnt7, M. musculus*) | GCACAGGTTTACGCACATCA |
| Sequence-based reagent | GalnT7-Rev | This paper | QPCR primer (amplify *Galnt7, M. musculus*) | TTCCAGGCGGTTTTCAGTCC |
| Sequence-based reagent | GalnT9-For | This paper | QPCR primer (amplify *Galnt9, M. musculus*) | CAACTTTGGGCTGCGGTTAG |
| Sequence-based reagent | GalnT9-Rev | This paper | QPCR primer (amplify *Galnt9, M. musculus*) | CCCACATTGCTCTTGGGTCT |
| Sequence-based reagent | GalnT10-For | This paper | QPCR primer (amplify *Galnt10, M. musculus*) | GGAGTACCGCCACCTCTCAG |
| Sequence-based reagent | GalnT10-Rev | This paper | QPCR primer (amplify *Galnt10, M. musculus*) | AGGTCCCAGGCAATTTTGGT |

*Continued on next page*

*Appendix 1—table 1 continued*

| Reagent type (species) or resource | Designation | Source or reference | Identifiers | Additional information |
|---|---|---|---|---|
| Sequence-based reagent | GalnT11-For | This paper | QPCR primer (amplify *Galnt11, M. musculus*) | GGCTGTACCAAGTGTCCGTT |
| Sequence-based reagent | GalnT11-Rev | This paper | QPCR primer (amplify *Galnt11, M. musculus*) | GCAGGCATGACAAAACCAGG |
| Sequence-based reagent | GalnT12-For | This paper | QPCR primer (amplify *Galnt12, M. musculus*) | ACAACGGCTTTGCACCATAC |
| Sequence-based reagent | GalnT12-Rev | This paper | QPCR primer (amplify *Galnt12, M. musculus*) | ACACTCTTGTGACACCCAGC |
| Sequence-based reagent | GalnT13-For | This paper | QPCR primer (amplify *Galnt13, M. musculus*) | CTGGCAATGTGGAGGTTCTT |
| Sequence-based reagent | GalnT13-Rev | This paper | QPCR primer (amplify *Galnt13, M. musculus*) | AATTCATCCATCCACACTTCTGC |
| Sequence-based reagent | GalnT14-For | This paper | QPCR primer (amplify *Galnt14, M. musculus*) | TCTTTCCGAGTGTGGATGTGT |
| Sequence-based reagent | GalnT14-Rev | This paper | QPCR primer (amplify *Galnt14, M. musculus*) | CCCATCGGGGAAAACATAAGGA |
| Sequence-based reagent | GalnT15-For | This paper | QPCR primer (amplify *Galnt15, M. musculus*) | CTGCGGTGGCTCTGTTGAAA |
| Sequence-based reagent | GalnT15-Rev | This paper | QPCR primer (amplify *Galnt15, M. musculus*) | CTGGGATGTGCCTGTAGAAGG |
| Sequence-based reagent | GalnT16-For | This paper | QPCR primer (amplify *Galnt16, M. musculus*) | TGGTGACCAGCAAATGTCAGA |
| Sequence-based reagent | GalnT16-Rev | This paper | QPCR primer (amplify *Galnt16, M. musculus*) | TCCGGTCGAAATGTGAGGAG |
| Sequence-based reagent | GalnT18-For | This paper | QPCR primer (amplify *Galnt18, M. musculus*) | CAGAAGTGCTCGGGACAACA |
| Sequence-based reagent | GalnT18-Rev | This paper | QPCR primer (amplify *Galnt18, M. musculus*) | TTGGCTCTCCCTCTCAGACT |
| Sequence-based reagent | Galntl5-For | This paper | QPCR primer (amplify *Galntl5, M. musculus*) | AGTGAGCGCGTGGAATTAAG |
| Sequence-based reagent | Galntl5-Rev | This paper | QPCR primer (amplify *Galntl5, M. musculus*) | AGATTTGTCCTGTGGTGCGA |
| Sequence-based reagent | Wbscr17-For | This paper | QPCR primer (amplify *Wbscr17, M. musculus*) | CTTAGGTGCTCTGGGGACCA |

*Continued on next page*

*Appendix 1—table 1 continued*

| Reagent type (species) or resource | Designation | Source or reference | Identifiers | Additional information |
|---|---|---|---|---|
| Sequence-based reagent | Wbscr17-Rev | This paper | QPCR primer (amplify *Wbscr17*, *M. musculus*) | TGTACAAGCTGCTCTTGACCT |
| Sequence-based reagent | Galntl6-For | This paper | QPCR primer (amplify *Galntl6*, *M. musculus*) | ACCGAGACTAGCAGTTCCCT |
| Sequence-based reagent | Galntl6-Rev | This paper | QPCR primer (amplify *Galntl6*, *M. musculus*) | GTCATGCGCTCTGTTTCCAC |
| Sequence-based reagent | Actin beta-For | This paper | QPCR primer (amplify *Actb*, *M. musculus*) | GACCTCTAT GCCAACACAGT |
| Sequence-based reagent | Actin beta-Rev | This paper | QPCR primer (amplify *Actb*, *M. musculus*) | AGTACTTGC GCTCAGGAGGA |
| Sequence-based reagent | Ins1- For | This paper | QPCR primer (amplify *Ins1*, R. Norvegicus) | ACCCTAAGTGACCAGCTACA |
| Sequence-based reagent | Ins1-Rev | This paper | QPCR primer (amplify *Ins1*, R. Norvegicus) | TTCACGACGGGACTTGGG |
| Sequence-based reagent | Gapdh-For | This paper | QPCR primer (amplify *Gapdh*, R. Norvegicus) | AGTGCCAGCCTCGTCTCATA |
| Sequence-based reagent | Gapdh-Rev | This paper | QPCR primer (amplify *Gapdh*, R. Norvegicus) | GATGGTGATGGGTTTCCCGT |

