## [Decision Letter]

**Acceptance summary:**

The study not only highlights the importance of *O*-glycosylation as a novel posttranslational modification, but also documents species specificity. Notably, mouse osteocalcin is *O*-glycosylated at serine 8, whereas human osteocalcin is not. Review comments have been thoughtfully and thoroughly addressed.

**Decision letter after peer review:**

Thank you for submitting your article "The half-life of the bone-derived hormone osteocalcin is regulated through *O*-glycosylation in mice, but not in humans" for consideration by *eLife*. Your article has been reviewed by three peer reviewers, and the evaluation has been overseen by Mone Zaidi as the Reviewing Editor and Clifford Rosen as the Senior Editor. The following individual involved in review of your submission has agreed to reveal their identity: T John Martin (Reviewer #1).

The reviewers have discussed the reviews with one another and the Reviewing Editor has drafted this decision to help you prepare a revised submission.

As the editors have judged that your manuscript is of interest, but as described below that additional experiments are required before we consider it for publication, we would like to draw your attention to changes in our revision policy that we have made in response to COVID-19 (https://elifesciences.org/articles/57162). First, because many researchers have temporarily lost access to the labs, we will give authors as much time as they need to submit revised manuscripts. We are also offering, if you choose, to post the manuscript to bioRxiv (if it is not already there) along with this decision letter and a formal designation that the manuscript is "in revision at *eLife*". Please let us know if you would like to pursue this option. (If your work is more suitable for medRxiv, you will need to post the preprint yourself, as the mechanisms for us to do so are still in development.)

Summary:

The work establishes that mouse osteocalcin is *O*-glycosylated at Ser-8, independently of processing and γ-carboxylation. Human osteocalcin was found not to be *O*-glycosylated, but mutation to Ser-8 allowed that process to occur. If increased stability of osteocalcin in vitro as a result of *O*-glycosylation is real, it could be of interest. The paper was found to be well written with figures illustrating the findings. With that said, several key experiments are required that would considerably strengthen the conclusions. Notably, without information on the biological outcomes of *O*-glycosylation, the paper is seen to be of limited interest.

Essential revisions:

1) Regarding the ELISA (Ferron et al., 2010b), capture antibodies can distinguish between carboxylated and non-carboxylated OCN. For the present work, it is essential that the authors show that OCN with or without *O*-glycosylation at Ser-8 is measured identically in this ELISA. The authors should specify what capture antibodies were used. This query applies also to data with human OCN and the Ser-8 mutant that is *O*-glycosylated. Furthermore, does the commercial ELISA kit measure glycosylated mutant and normal hOCN as identical?

2) Figure 3H-J show a statistically significant difference in levels of glycosylated and non-glycosylated mouse OCN. These experiments however do not measure "half-life" as claimed in the title and Abstract. The in vivo half-life of injected *O*-glycosylated vs. wt ucOCN should therefore be compared using timed estimations during the declining phase.

3) A feature of OCN that interested the authors was the remarkable difference in circulating amounts – more than 10 times higher in the mouse. This work appears to be part of a search for mechanisms to explain this, although they might consider that these are evolutionary changes, including the fact that there is only 65 % conservation of sequence between mouse and human , and the human OCN is not *O*-glycosylated, whereas the mouse OCN is. The biological significance of this difference in *O*-glycosylation thus needs to be established. While knocking-in a mutation to abolish *O*-glycosylation will provide the most definitive answer, the reviewers consider this not to be feasible during the pandemic. Therefore, at the very least, a cell-based assay should be used to compare biological activity. Examples are the dose-dependent increase in insulin mRNA in mouse pancreatic islets (Ferron. et al., PNAS 105: 5266, 2008). Very low dose ucOCN was shown in that work to promote insulin expression in islets in a dose-dependent manner. An alternative approach, arising from the same paper, would be to show at slightly higher doses, a dose-dependent increase in adiponectin expression in mouse adipocytes. The authors might have other possibilities of cell- based assay.

---

## [Author Response]

Essential revisions:1) Regarding the ELISA (Ferron et al., 2010b), capture antibodies can distinguish between carboxylated and non-carboxylated OCN. For the present work, it is essential that the authors show that OCN with or without O-glycosylation at Ser-8 is measured identically in this ELISA.

The reviewers raised an important point. We now show in Figure 3—figure supplement 2 a standard curve of non-glycosylated non-carboxylated mouse OCN (ucOCN) ran side-by-side with a standard curve of *O*-glycosylated non-carboxylated mouse OCN (*O*-gly ucOCN). It shows that our total mouse osteocalcin ELISA assay reacts equally with *O-*glycosylated and non-glycosylated mouse OCN at concentrations ranging from 12.5 to 50 ng/ml. However, the assay slightly underestimates the *O*-gly ucOCN at 100 ng/ml compared to non-glycosylated OCN. For this reason, in all our experiments, samples were diluted at concentration of less than 50 ng/ml prior to be measured in the ELISA. This is now clearly stated in the Materials and methods section, and in the Results section.

The authors should specify what capture antibodies were used.

As detailed in the original paper describing this ELISA assay (Ferron et al., 2010), the capture antibody is a goat polyclonal antibody directed against the central part (amino acids 11 to 26) of mouse OCN (anti-MID OCN) and recognizing with equal affinity the uncarboxylated and carboxylated OCN proteins. The detection antibody is a goat polyclonal antibody directed against the C-terminal region (amino acids 26 to 46) of mouse OCN (anti-CTERM OCN). Of note, the glycosylation site (Ser 8) is not comprised in these two epitopes, most likely explaining why the ELISA recognizes both forms. The information on the capture and detection antibodies used has now been added to the Materials and methods section.

This query applies also to data with human OCN and the Ser-8 mutant that is O-glycosylated.

As detailed in the publication reporting the human ucOCN ELISA (Lacombe et al., 2020), the capture antibody in this assay is a mouse monoclonal antibody (8H4) specific to the C-terminal region of human OCN (i.e., amino acids 30 to 49). The detection antibody is a mouse monoclonal antibody (4B6) specific to the mid-region of human ucOCN (i.e., amino acids 12 to 28 in ucOCN). This information has now been added to the Materials and methods section.

Furthermore, does the commercial ELISA kit measure glycosylated mutant and normal hOCN as identical?

To answer this valid question, we now show in Figure 5—figure supplement 2 the standard curve of non-glycosylated non-carboxylated human OCN (uc-hOCN) ran side-by-side with a standard curve of *O*-glycosylated non-carboxylated human OCN (*O*-gly uc-hOCN). This shows that the human uncarboxylated OCN ELISA assay recognizes equally glycosylated and non-glycosylated human OCN at all the concentrations tested (i.e., 0.2-6.4 ng/ml). This is now stated in the Results section, and in the Materials and methods section.

2) Figure 3H-J show a statistically significant difference in levels of glycosylated and non-glycosylated mouse OCN. These experiments however do not measure "half-life" as claimed in the title and Abstract. The in vivo half-life of injected O-glycosylated vs. wt ucOCN should therefore be compared using timed estimations during the declining phase.

To answer this point, we reanalyzed our data on OCN in vivo stability. As suggested by the reviewer, we calculated the % of OCN in serum relative to the peak at 30 minutes. This new result, now presented in Figure 3I, shows that glycosylated ucOCN has a significantly longer half-life in circulation compared to non-glycosylated form in fasting conditions (i.e., ~182 vs. ~108 minutes). These new data are presented in Figure 3I and mentioned in the text on page 13, line 277.

3) A feature of OCN that interested the authors was the remarkable difference in circulating amounts – more than 10 times higher in the mouse. This work appears to be part of a search for mechanisms to explain this, although they might consider that these are evolutionary changes, including the fact that there is only 65 % conservation of sequence between mouse and human , and the human OCN is not O-glycosylated, whereas the mouse OCN is. The biological significance of this difference in O-glycosylation thus needs to be established. While knocking-in a mutation to abolish O-glycosylation will provide the most definitive answer, the reviewers consider this not to be feasible during the pandemic. Therefore, at the very least, a cell-based assay should be used to compare biological activity. Examples are the dose-dependent increase in insulin mRNA in mouse pancreatic islets (Ferron. et al., PNAS 105: 5266, 2008). Very low dose ucOCN was shown in that work to promote insulin expression in islets in a dose-dependent manner. An alternative approach, arising from the same paper, would be to show at slightly higher doses, a dose-dependent increase in adiponectin expression in mouse adipocytes. The authors might have other possibilities of cell- based assay.

We thank the reviewers for this suggestion. To address it, we have established a cell-based assay using the INS-1 832/3 rat insulinoma cell line, which respond well to natural incretin hormones such as glucagon-like peptide 1 and gastric inhibitory peptide (Ronnebaum et al., 2008). In the condition of this assay, we found that, compared to vehicle, low doses of mouse *O-*gly ucOCN (0.3 and 0.1 ng/ml) can increase the expression of the insulin encoding gene, *Ins1*, suggesting that *O*-gly ucOCN is biologically active. Moreover, although 0.3 ng/ml of non-glycosylated ucOCN was also able to increase *Ins1* expression by ~1.5-fold, we noted that the *O-*gly ucOCN had a stronger effect at similar or even lower doses (new Figure 4A). This difference in the biological activity in culture could be explained at least in part by the observation that *O-*gly ucOCN is more stable than ucOCN when incubated in INS-1 832/3 cell cultures (see new Figure 4B). Overall, these new data support the notion that *O-*gly ucOCN is biologically active, at least in a cell-based assay, and that *O*-glycosylation also increases the stability of ucOCN in cell culture. These new results are included in Figure 4A and B, and verbalized in the subsection “Impact of glycosylation on OCN activity in culture”. They are also discussed in the Discussion section.